# Nonparametric estimation of continuous DPPs with kernel methods

**Michaël Fanuel and Rémi Bardenet**
Université de Lille, CNRS, Centrale Lille
UMR 9189 – CRIStAL, F-59000 Lille, France
{michael.fanuel, remi.bardenet}@univ-lille.fr

## Abstract

Determinantal Point Process (DPPs) are statistical models for repulsive point patterns. Both sampling and inference are tractable for DPPs, a rare feature among models with negative dependence that explains their popularity in machine learning and spatial statistics. Parametric and nonparametric inference methods have been proposed in the finite case, i.e. when the point patterns live in a finite ground set. In the continuous case, only parametric methods have been investigated, while nonparametric maximum likelihood for DPPs – an optimization problem over trace-class operators – has remained an open question. In this paper, we show that a restricted version of this maximum likelihood (MLE) problem falls within the scope of a recent representer theorem for nonnegative functions in an RKHS. This leads to a finite-dimensional problem, with strong statistical ties to the original MLE. Moreover, we propose, analyze, and demonstrate a fixed point algorithm to solve this finite-dimensional problem. Finally, we also provide a controlled estimate of the correlation kernel of the DPP, thus providing more interpretability.

## 1 Introduction

Determinantal point processes (DPPs) are a tractable family of models for repulsive point patterns, where interaction between points is parametrized by a positive semi-definite kernel. They were introduced by Macchi [1975] in the context of fermionic optics, and have gained a lot of interest since the 2000s, in particular in probability [Hough et al., 2006], spatial statistics [Lavancier et al., 2014], and machine learning [Kulesza and Taskar, 2012]. In machine learning at large, DPPs have been used essentially for two purposes: as statistical models for diverse subsets of items, like in recommendation systems [Gartrell et al., 2019], and as subsampling tools, like in experimental design [Derezinski et al., 2020], column subset selection [Belhadji et al., 2020b], or Monte Carlo integration [Gautier et al., 2019; Belhadji et al., 2019]. In this paper, we are concerned with DPPs used as statistical models for repulsion, and more specifically with inference for continuous DPPs.

DPP models in Machine Learning (ML) have so far mostly been *finite* DPPs: they are distributions over subsets of a (large) finite ground set, like subsets of sentences from a large corpus of documents [Kulesza and Taskar, 2012]. Since Affandi et al. [2014], a lot of effort has been put into designing efficient inference procedures for finite DPPs. In particular, the fixed point algorithm of Mariet and Sra [2015] allows for nonparametric inference of a finite DPP kernel, thus learning the features used for modelling diversity from the data. DPP models on infinite ground sets, say $\mathbb{R}^d$, while mathematically and algorithmically very similar to finite DPPs, have been less popular in ML than in spatial statistics. It is thus natural that work on inference for *continuous* DPPs has happened mostly in the latter community; see e.g. the seminal paper [Lavancier et al., 2014]. Inference for continuous DPPs has however focused on the parametric setting, where a handful of interpretable parameters are learned. Relatedly, spatial statisticians typically learn the *correlation* kernel of a DPP, which is more

35th Conference on Neural Information Processing Systems (NeurIPS 2021).

interpretable, while machine learners focus on the *likelihood* kernel, with structural assumptions to make learning scale to large ground sets.

In this paper, we tackle *nonparametric* inference for continuous DPPs using recent results on kernel methods. More precisely, maximum likelihood estimation (MLE) for continuous DPPs is an optimization problem over trace-class operators. Our first contribution is to show that a suitable modification of this problem is amenable to the representer theorem of Marteau-Ferey, Bach, and Rudi [2020]. Further drawing inspiration from the follow-up work [Rudi, Marteau-Ferey, and Bach, 2020], we derive an optimization problem over matrices, and we prove that its solution has a near optimal objective in the original MLE problem. We then propose, analyze, and demonstrate a fixed point algorithm for the resulting finite problem, in the spirit [Mariet and Sra, 2015] of nonparametric inference for finite DPPs. While our optimization pipeline focuses on the so-called likelihood kernel of a DPP, we also provide a controlled sampling approximation to its correlation kernel, thus providing more interpretability of our estimated kernel operator. A by-product contribution of independent interest is an analysis of a sampling approximation for Fredholm determinants.

The rest of the paper is organized as follows. Since the paper is notation-heavy, we first summarize our notation and give standard definitions in Section 1.1. In Section 2, we introduce DPPs and prior work on inference. In Section 3, we introduce our constrained maximum likelihood problem, and study its empirical counterpart. We analyze an algorithm to solve the latter in Section 4. Statistical guarantees are stated in Section 5, while Section 6 is devoted to numerically validating the whole pipeline. Our code is freely available[1].

## 1.1 Notation and background

**Sets.** It is customary to define DPPs on a compact Polish space $\mathcal{X}$ endowed with a Radon measure $\mu$, so that we can define the space of square integrable functions $L^2(\mathcal{X})$ for this measure [Hough et al., 2009]. Outside of generalities in Section 2, we consider a compact $\mathcal{X} \subset \mathbb{R}^d$ and $\mu$ the uniform probability measure on $\mathcal{X}$. Let $(\mathcal{H}, \langle \cdot, \cdot \rangle)$ be an RKHS of functions on $\mathcal{X}$ with a bounded continuous kernel $k_{\mathcal{H}}(x, y)$, and let $\kappa^2 = \sup_{x \in \mathcal{X}} k_{\mathcal{H}}(x, x)$. Denote by $\phi(x) = k_{\mathcal{H}}(x, \cdot) \in \mathcal{H}$ the canonical feature map. For a Hilbert space $\mathcal{F}$, denote by $\mathcal{S}_+(\mathcal{F})$ the space of symmetric and positive semi-definite trace-class operators on $\mathcal{F}$. By a slight abuse of notation, we denote by $\mathcal{S}_+(\mathbb{R}^n)$ the space of $n \times n$ real positive semi-definite matrices. Finally, all sets are denoted by calligraphic letters (e.g. $\mathcal{C}, \mathcal{I}$).

**Operators and matrices.** Trace-class endomorphisms of $L^2(\mathcal{X})$, seen as integral operators, are typeset as uppercase sans-serif (e.g. A, K), and the corresponding integral kernels as lowercase sans-serif (e.g. a, k). Notice that $\mathsf{k}(x, y)$ and $k_{\mathcal{H}}(x, y)$ are distinct functions. Other operators are written in standard fonts (e.g. $A, S$), while we write matrices and finite-dimensional vectors in bold (e.g. $\mathbf{K}, \mathbf{C}, \mathbf{v}$). The identity operator is written commonly as $\mathbb{I}$, whereas the $n \times n$ identity matrix is denoted by $\mathbf{I}_n$. When $\mathcal{C}$ is a subset of $\{1, \ldots, n\}$ and $\mathbf{K}$ is an $n \times n$ matrix, the matrix $\mathbf{K}_{\mathcal{C}\mathcal{C}}$ is the square submatrix obtained by selecting the rows and columns of $\mathbf{K}$ indexed by $\mathcal{C}$.

**Restriction and reconstruction operators.** Following Rosasco et al. [2010, Section 3], we define the restriction operator $S : \mathcal{H} \to L^2(\mathcal{X})$ as $(Sg)(x) = g(x)$. Its adjoint $S^* : L^2(\mathcal{X}) \to \mathcal{H}$ is the reconstruction operator $S^*h = \int_{\mathcal{X}} h(x)\phi(x)\mathrm{d}\mu(x)$. The classical integral operator given by $\mathsf{T}_{k_{\mathcal{H}}}h = \int_{\mathcal{X}} k_{\mathcal{H}}(\cdot, x)h(x)\mathrm{d}\mu(x)$, seen as an endomorphism of $L^2(\mathcal{X})$, thus takes the simple expression $\mathsf{T}_{k_{\mathcal{H}}} = SS^*$. Similarly, the so-called covariance operator $C : \mathcal{H} \to \mathcal{H}$, defined by $C = \int_{\mathcal{X}} \phi(x) \otimes \overline{\phi(x)}\mathrm{d}\mu(x)$, writes $C = S^*S$. In the tensor product notation defining $C$, $\overline{\phi(x)}$ is an element of the dual of $\mathcal{H}$ and $\phi(x) \otimes \overline{\phi(x)}$ is the endomorphism of $\mathcal{H}$ defined by $((\phi(x) \otimes \overline{\phi(x)})(g) = g(x)\phi(x)$; see e.g. Sterge et al. [2020]. Finally, for convenience, given a finite set $\{x_1, \ldots, x_n\} \subset \mathcal{X}$, we also define *discrete* restriction and reconstruction operators, respectively, as $S_n : \mathcal{H} \to \mathbb{R}^n$ such that $S_n g = (1/\sqrt{n})[g(x_1), \ldots, g(x_n)]^\top$, and $S_n^* \mathbf{v} = (1/\sqrt{n}) \sum_{i=1}^n \mathbf{v}_i \phi(x_i)$ for any $\mathbf{v} \in \mathbb{R}^n$. In particular, we have $S_n S_n^* = (1/n)\mathbf{K}$ where $\mathbf{K} = [k_{\mathcal{H}}(x_i, x_j)]_{1 \leq i,j \leq n}$ is a kernel matrix, which is defined for a given ordering of the set $\{x_1, \ldots, x_n\}$. To avoid cumbersome expressions, when several discrete sets of different cardinalities, say $n$ and $p$, are used, we simply write the respective sampling operators as $S_n$ and $S_p$.

---

[1] https://github.com/mrfanuel/LearningContinuousDPPs.jl

## 2 Determinantal point processes and inference

**Determinantal point processes and L-ensembles.** Consider a simple point process $\mathcal{Y}$ on $\mathcal{X}$, that is, a random discrete subset of $\mathcal{X}$. For $\mathcal{D} \subset \mathcal{X}$, we denote by $\mathcal{Y}(\mathcal{D})$ the number of points of this process that fall within $\mathcal{D}$. Letting $m$ be a positive integer, we say that $\mathcal{X}$ has $m$-point correlation function $\varrho_m$ w.r.t. to the reference measure $\mu$ if, for any mutually disjoint subsets $\mathcal{D}_1, \ldots, \mathcal{D}_m \subset \mathcal{X}$,

$$\mathbb{E}\left[\prod_{i=1}^{m} \mathcal{Y}(\mathcal{D}_i)\right] = \int_{\prod_{i=1}^{m} \mathcal{D}_i} \varrho_m(x_1, \ldots, x_m) \mathrm{d}\mu(x_1) \ldots \mathrm{d}\mu(x_m).$$

In most cases, a point process is characterized by its correlation functions $(\rho_m)_{m \geq 1}$. In particular, a determinantal point process (DPP) is defined as having correlation functions in the form of a determinant of a Gram matrix, i.e. $\varrho_m(x_1, \ldots, x_m) = \det[\mathsf{k}(x_i, x_j)]$ for all $m \geq 1$. We then say that $\mathsf{k}$ is the *correlation kernel* of the DPP. Not all kernels yield a DPP: if $\mathsf{k}(x, y)$ is the integral kernel of an operator $\mathsf{K} \in \mathcal{S}_+(L^2(\mathcal{X}))$, the Macchi-Soshnikov theorem [Macchi, 1975; Soshnikov, 2000] states that the corresponding DPP exists if and only if the eigenvalues of $\mathsf{K}$ are within $[0, 1]$. In particular, for a finite ground set $\mathcal{X}$, taking the reference measure to be the counting measure leads to conditions on the kernel matrix; see Kulesza and Taskar [2012].

A particular class of DPPs is formed by the so-called L-ensembles, for which the correlation kernel writes

$$\mathsf{K} = \mathsf{A}(\mathsf{A} + \mathbb{I})^{-1}, \tag{1}$$

with the *likelihood* operator $\mathsf{A} \in \mathcal{S}_+(L^2(\mathcal{X}))$ taken to be of the form $\mathsf{A}f(x) = \int_{\mathcal{X}} \mathsf{a}(x, y)f(y)\mathrm{d}\mu(y)$. The kernel $\mathsf{a}$ of $\mathsf{A}$ is sometimes called the *likelihood kernel* of the L-ensemble, to distinguish it from its correlation kernel $\mathsf{k}$. The interest of L-ensembles is that their Janossy densities can be computed in closed form. Informally, the $m$-Janossy density describes the probability that the point process has cardinality $m$, and that the points are located around a given set of distinct points $x_1, \ldots, x_m \in \mathcal{X}$. For the rest of the paper, we assume that $\mathcal{X} \subset \mathbb{R}^d$ is compact, and that $\mu$ is the uniform probability measure on $\mathcal{X}$; our results straightforwardly extend to other densities w.r.t. Lebesgue. With these assumptions, the $m$-Janossy density is proportional to

$$\det(\mathbb{I} + \mathsf{A})^{-1} \cdot \det[\mathsf{a}(x_i, x_j)]_{1 \leq i,j \leq m}, \tag{2}$$

where the normalization constant is a Fredholm deteminant that implicitly depends on $\mathcal{X}$. Assume now that we are given $s$ i.i.d. samples of a DPP, denoted by the sets $\mathcal{C}_1, \ldots, \mathcal{C}_s$ in the window $\mathcal{X}$. The associated maximum log-likelihood estimation problem reads

$$\max_{\mathsf{A} \in \mathcal{S}_+(L^2(\mathcal{X}))} \frac{1}{s} \sum_{\ell=1}^{s} \log \det \left[\mathsf{a}(x_i, x_j)\right]_{i,j \in \mathcal{C}_\ell} - \log \det(\mathbb{I} + \mathsf{A}). \tag{3}$$

Solving (3) is a nontrivial problem. First, it is difficult to calculate the Fredholm determinant. Second, it is not straightforward to optimize over the space of operators $\mathcal{S}_+(L^2(X))$ in a nonparametric setting. However, we shall see that the problem becomes tractable if we restrict the domain of (3) and impose regularity assumptions on the integral kernel $\mathsf{a}(x, y)$ of the operator $\mathsf{A}$. For more details on DPPs, we refer the interested reader to Hough et al. [2006]; Kulesza and Taskar [2012]; Lavancier et al. [2014].

**Previous work on learning DPPs.** While continuous DPPs have been used in ML as sampling tools [Belhadji et al., 2019, 2020a] or models [Bardenet and Titsias, 2015; Ghosh and Rigollet, 2020], their systematic parametric estimation has been the work of spatial statisticians; see Lavancier et al. [2015]; Biscio and Lavancier [2017]; Poinas and Lavancier [2021] for general parametric estimation through (3) or so-called *minimum-contrast* inference. Still for the parametric case, a two-step estimation was recently proposed for non-stationary processes by Lavancier et al. [2021]. In a more general context, non-asymptotic risk bounds for estimating a DPP density are given in Baraud [2013].

Discrete DPPs have been more common in ML, and the study of their estimation has started some years ago [Affandi et al., 2014]. Unlike continuous DPPs, nonparametric estimation procedures have been investigated for finite DPPs by Mariet and Sra [2015], who proposed a fixed point algorithm. Moreover, the statistical properties of maximum likelihood estimation of discrete L-ensembles were studied by Brunel et al. [2017b]. We can also cite low-rank approaches [Dupuy and Bach, 2018;

Gartrell et al., 2017], learning with negative sampling [Mariet et al., 2019], learning with moments and cycles [Urschel et al., 2017], or learning with Wasserstein training [Anquetil et al., 2020]. Learning non-symmetric finite DPPs [Gartrell et al., 2019, 2021] has also been proposed, motivated by recommender systems.

At a high level, our paper is a continuous counterpart to the nonparametric learning of finite DPPs with symmetric kernels in Mariet and Sra [2015]. Our treatment of the continuous case is made possible by recent advances in kernel methods .

## 3 A sampling approximation to a constrained MLE

Using the machinery of kernel methods, we develop a controlled approximation of the MLE problem (3). Let us outline the main landmarks of our approach. First, we restrict the domain of the MLE problem (3) to smooth operators. On the one hand, this restriction allows us to develop a sampling approximation of the Fredholm determinant. On the other hand, the new optimization problem now admits a finite rank solution that can be obtained by solving a finite-dimensional problem. This procedure is described in Algorithm 1 and yields an estimator for the likelihood kernel. Finally, we use another sampling approximation and solve a linear system to estimate the correlation kernel of the fitted DPP; see Algorithm 2.

**Restricting to smooth operators.** In order to later apply the representer theorem of Marteau-Ferey et al. [2020], we restrict the original maximum likelihood problem (3) to "smooth" operators $\mathsf{A} = SAS^*$, with $A \in \mathcal{S}_+(\mathcal{H})$ and $S$ the restriction operator introduced in Section 1.1. Note that the kernel of $\mathsf{A}$ now writes

$$\mathsf{a}(x,y) = \langle \phi(x), A\phi(y) \rangle. \tag{4}$$

With this restriction on its domain, the optimization problem (3) now reads

$$\min_{A \in \mathcal{S}_+(\mathcal{H})} f(A) = -\frac{1}{s} \sum_{\ell=1}^{s} \log \det \left[ \langle \phi(x_i), A\phi(x_j) \rangle \right]_{i,j \in \mathcal{C}_\ell} + \log \det(\mathbb{I} + SAS^*). \tag{5}$$

**Approximating the Fredholm determinant.** We use a sampling approach to approximate the normalization constant in (5). We sample a set of points $\mathcal{I} = \{x_i' : 1 \le i \le n\}$ i.i.d. from the ambient probability measure $\mu$. For definiteness, we place ourselves on an event happening with probability one where all the points in $\mathcal{I}$ and $\mathcal{C}_\ell$ for $1 \le \ell \le s$ are distinct. We define the sample version of $f(A)$ as

$$f_n(A) = -\frac{1}{s} \sum_{\ell=1}^{s} \log \det \left[ \langle \phi(x_i), A\phi(x_j) \rangle \right]_{i,j \in \mathcal{C}_\ell} + \log \det(\mathbf{I}_n + S_n A S_n^*),$$

where the Fredholm determinant of $\mathsf{A} = SAS^*$ has been replaced by the determinant of a *finite* matrix involving $S_n A S_n^* = [\langle \phi(x_i'), A\phi(x_j') \rangle]_{1 \le i,j \le n}$.

**Theorem 1** (Approximation of the Fredholm determinant). *Let $\delta \in (0, 1/2)$. With probability at least $1 - 2\delta$,*

$$|\log \det(\mathbf{I}_n + S_n A S_n^*) - \log \det(\mathbb{I} + SAS^*)| \le \log \det(\mathbb{I} + c_n A),$$

*with*

$$c_n = \frac{4\kappa^2 \log \left( \frac{2\kappa^2}{\ell\delta} \right)}{3n} + \sqrt{\frac{2\kappa^2 \ell \log \left( \frac{2\kappa^2}{\ell\delta} \right)}{n}},$$

*where $\ell = \lambda_{\max}(\mathsf{T}_{k_{\mathcal{H}}})$ and $\kappa^2 = \sup_{x \in \mathcal{X}} k_{\mathcal{H}}(x,x) < \infty$.*

The proof of Theorem 1 is given in Supplementary Material in Section S3.2. Several remarks are in order. First, the high probability[2] in the statement of Theorem 1 is that of the event $\{\|S^*S - S_n^* S_n\|_{op} \lesssim c_n\}$. Importantly, all the results given in what follows for the approximation of the solution of (5) only depend on this event, so that we do not need any union bound. Second, we emphasize that $\mathsf{T}_{k_{\mathcal{H}}}$, defined in Section 1.1, should not be confused with the correlation kernel (1). Third, to interpret the bound in Theorem 1, it is useful to recall that $\log \det(\mathbb{I} + c_n A) \le c_n \operatorname{Tr}(A)$,

---

[2]We write $a \lesssim b$ if there exists a constant $c > 0$ such that $a \le cb$.

since $A \in \mathcal{S}_+(\mathcal{H})$. Thus, by penalizing $\mathrm{Tr}(A)$, one also improves the upper bound on the Fredholm determinant approximation error. This remark motivates the following infinite dimensional problem

$$\min_{A \in \mathcal{S}_+(\mathcal{H})} f_n(A) + \lambda \mathrm{Tr}(A), \tag{6}$$

for some $\lambda > 0$. The penalty on $\mathrm{Tr}(A)$ is also intuitively needed so that the optimization problem selects a smooth solution, i.e., such a trace regularizer promotes a fast decay of eigenvalues of $A$. Note that this problem depends both on the data $\mathcal{C}_1, \ldots, \mathcal{C}_n \subset \mathcal{X}$ and the subset $\mathcal{I}$ used for approximating the Fredholm determinant.

**Finite-dimensional representatives.** In an RHKS, there is a natural mapping between finite rank operators and matrices. For the sake of completeness, let $\mathbf{K} = [k_\mathcal{H}(z_i, z_j)]_{1 \le i,j \le m}$ be a kernel matrix and let $\mathbf{K} = \mathbf{R}^\top \mathbf{R}$ be a Cholesky factorization. Throughout the paper, kernel matrices are always assumed to be invertible. This is not a strong assumption: if $k_\mathcal{H}$ is the Laplace, Gaussian or Sobolev kernel, this is true almost surely if $z_i$ for $1 \le i \le m$ are sampled e.g. w.r.t. the Lebesgue measure; see Bochner's classical theorem [Wendland, 2004, Theorem 6.6 and Corollary 6.9]. In this case, we can define a partial isometry $V : \mathcal{H} \to \mathbb{R}^m$ as $V = \sqrt{m}(\mathbf{R}^{-1})^\top S_m$. It satisfies $VV^* = \mathbf{I}$, and $V^*V$ is the orthogonal projector onto the span of $\phi(z_i)$ for all $1 \le i \le m$. This is helpful to define

$$\mathbf{\Phi}_i = V\phi(z_i) \in \mathbb{R}^n, \tag{7}$$

the finite-dimensional representative of $\phi(z_i) \in \mathcal{H}$ for all $1 \le i \le m$. This construction yields a useful mapping between an operator in $\mathcal{S}_+(\mathcal{H})$ and a finite matrix, which is instrumental for obtaining our results.

**Lemma 2** (Finite dimensional representatives, extension of Lemma 3 in Rudi et al. [2020]). *Let $A \in \mathcal{S}_+(\mathcal{H})$. Then, the matrix $\bar{\mathbf{B}} = VAV^*$ is such that $\mathbf{\Phi}_i^\top \bar{\mathbf{B}} \mathbf{\Phi}_j = \langle \phi(z_i), A\phi(z_j) \rangle$ for all $1 \le i, j \le m$, and $\log\det(\mathbf{I} + \bar{\mathbf{B}}) \le \log\det(\mathbb{I} + A)$, as well as $\mathrm{Tr}(\bar{\mathbf{B}}) \le \mathrm{Tr}(A)$.*

The proof of Lemma 2 is given in Section S3.1. Notice that the partial isometry $V$ also helps to map a matrix in $\mathcal{S}_+(\mathbb{R}^m)$ to an operator in $\mathcal{S}_+(\mathcal{H})$, as $\mathbf{B} \mapsto V^*\mathbf{B}V$, in such a way that we have the matrix element matching $\langle \phi(z_i), V^*\mathbf{B}V\phi(z_j) \rangle = \mathbf{\Phi}_i^\top \mathbf{B} \mathbf{\Phi}_j$ for all $1 \le i, j \le m$.

**Finite rank solution thanks to a representer theorem.** The sampling approximation of the Fredholm determinant also yields a finite rank solution for (6). For simplicity, we define $\mathcal{C} \triangleq \cup_{\ell=1}^s \mathcal{C}_\ell$ and recall $\mathcal{I} = \{x'_1, \ldots, x'_n\}$. Then, write the set of points $\mathcal{Z} \triangleq \mathcal{C} \cup \mathcal{I}$ as $\{z_1, \ldots, z_m\}$, with $m = |\mathcal{C}| + n$, and denote the corresponding restriction operator $S_m : \mathcal{H} \to \mathbb{R}^m$. Consider the kernel matrix $\mathbf{K} = [k_\mathcal{H}(z_i, z_j)]_{1 \le i,j \le m}$. In particular, since we used a trace regularizer, the representer theorem of Marteau-Ferey et al. [2020, Theorem 1] holds: the optimal operator is of the form

$$A = \sum_{i,j=1}^m \mathbf{C}_{ij} \phi(z_i) \otimes \overline{\phi(z_j)} \text{ with } \mathbf{C} \in \mathcal{S}_+(\mathbb{R}^m). \tag{8}$$

In this paper, we call $\mathbf{C}$ the representer matrix of the operator $A$. If we do the change of variables $\mathbf{B} = \mathbf{R}\mathbf{C}\mathbf{R}^\top$, we have the following identities: $A = mS_m^*\mathbf{C}S_m^* = V^*\mathbf{B}V$ and $\mathrm{Tr}(A) = \mathrm{Tr}(\mathbf{K}\mathbf{C}) = \mathrm{Tr}(\mathbf{B})$, thanks to Lemma S1 in Supplementary Material. Therefore, the problem (6) boils down to the *finite* non-convex problem:

$$\min_{\mathbf{B} \succeq 0} f_n(V^*\mathbf{B}V) + \lambda \mathrm{Tr}(\mathbf{B}), \tag{9}$$

where $f_n(V^*\mathbf{B}V) = -\frac{1}{s}\sum_{\ell=1}^s \log\det\left[\mathbf{\Phi}_i^\top \mathbf{B} \mathbf{\Phi}_j\right]_{i,j \in \mathcal{C}_\ell} + \log\det\left[\delta_{ij} + \mathbf{\Phi}_i^\top \mathbf{B} \mathbf{\Phi}_j/|\mathcal{I}|\right]_{i,j \in \mathcal{I}}$. We assume that there is a global minimizer of (9) that we denote by $\mathbf{B}_\star$. The final estimator of the integral kernel of the likelihood A depends on $\mathbf{C}_\star = \mathbf{R}^{-1}\mathbf{B}_\star\mathbf{R}^{-1\top}$ and reads $\hat{\mathsf{a}}(x, y) = \sum_{i,j=1}^m \mathbf{C}_{\star ij} k_\mathcal{H}(z_i, x) k_\mathcal{H}(z_j, y)$. The numerical strategy is summarized in Algorithm 1.

---

**Algorithm 1** Estimation of the integral kernel $\mathsf{a}(x, y)$ of the DPP likelihood kernel A.

---

**procedure** ESTIMATEA$(\lambda, \mathcal{C}_1, \ldots, \mathcal{C}_s)$
    Sample $\mathcal{I} = \{x'_1, \ldots, x'_n\}$ i.i.d. from $\mu$              ▷ Sample $n$ points for Fredhom det. approx.
    Define $\mathcal{Z} \triangleq \cup_{\ell=1}^s \mathcal{C}_\ell \cup \mathcal{I}$                               ▷ Collect all samples
    Compute $\mathbf{K} = \mathbf{R}^\top \mathbf{R}$ with $\mathbf{K} = [k_\mathcal{H}(z_i, z_j)]_{i,j \in \mathcal{Z}}$           ▷ Cholesky of kernel matrix
    Solve (9) with iteration (14) to obtain $\mathbf{B}_\star$               ▷ Regularized Picard iteration
    Compute $\mathbf{C}_\star = \mathbf{R}^{-1} \mathbf{B}_\star \mathbf{R}^{-1\top}$               ▷ Representer matrix of $\hat{\mathsf{a}}(x, y)$
    **return** $\hat{\mathsf{a}}(x, y) = \sum_{i,j=1}^m \mathbf{C}_{\star ij} k_\mathcal{H}(z_i, x) k_\mathcal{H}(z_j, y)$          ▷ Likelihood kernel
**end procedure**

---

**Estimation of the correlation kernel.** The exact computation of the correlation kernel of the L-ensemble DPP

$$\mathsf{K}(\gamma) = \mathsf{A}(\mathsf{A} + \gamma \mathbb{I})^{-1}, \tag{10}$$

requires the exact diagonalization of $\mathsf{A} = SAS^*$. For more flexibility, we introduced a scale parameter $\gamma > 0$ which often takes the value $\gamma = 1$. It is instructive to approximate $\mathsf{K}$ in order to easily express the correlation functions of the estimated point process. We propose here an approximation scheme based once again on sampling. Recall the form of the solution $A = mS_m^* \mathbf{C} S_m$ of (6), and consider the factorization $\mathbf{C} = \boldsymbol{\Lambda}^\top \boldsymbol{\Lambda}$ with $\boldsymbol{\Lambda} = \mathbf{F} \mathbf{R}^{-1\top}$ where $\mathbf{F}^\top \mathbf{F} = \mathbf{B}_\star$ is the Cholesky factorization of $\mathbf{B}_\star$. Let $\{x''_1, \ldots, x''_p\} \subseteq \mathcal{X}$ be sampled i.i.d. from the probability measure $\mu$ and denote by $S_p : \mathcal{H} \to \mathbb{R}^p$ the corresponding restriction operator. The following integral operator

$$\hat{\mathsf{K}} = mSS_m^* \boldsymbol{\Lambda}^\top (m\boldsymbol{\Lambda} S_m S_p^* S_p S_m^* \boldsymbol{\Lambda}^\top + \gamma \mathbf{I}_m)^{-1} \boldsymbol{\Lambda} S_m S^*, \tag{11}$$

gives an approximation of $\mathsf{K}$. The numerical approach for solving (11) relies on the computation of $\mathbf{K}_{mp} = \sqrt{mp} S_m S_p^* = [k_\mathcal{H}(z_i, x''_j)]$ with $1 \le i \le m$ and $1 \le j \le p$ is a rectangular kernel matrix, associated to a fixed ordering of $\mathcal{Z} = \{z_1, \ldots, z_m\}$ and $\{x''_1, \ldots, x''_p\}$. Our strategy is described in Algorithm 2.

---

**Algorithm 2** Estimation of the integral kernel $\mathsf{k}(x, y)$ of the DPP correlation kernel $\mathsf{K} = \mathsf{A}(\mathsf{A} + \mathbb{I})^{-1}$.

---

**procedure** ESTIMATEK$(\mathcal{Z}, \mathbf{C}_\star)$
    Compute $\mathbf{C}_\star = \boldsymbol{\Lambda}^\top \boldsymbol{\Lambda}$                       ▷ Factorization of representer matrix
    Sample $\{x''_1, \ldots, x''_p\} \subseteq \mathcal{X}$ i.i.d. from $\mu$                ▷ Sample $p$ points
    Compute $\mathbf{K}_{mp} = [k_\mathcal{H}(z_i, x''_j)] \in \mathbb{R}^{m \times p}$             ▷ Cross kernel matrix
    Compute $\boldsymbol{\Omega} = \boldsymbol{\Lambda}^\top (\boldsymbol{\Lambda} \mathbf{K}_{mp} \frac{1}{p} \mathbf{K}_{mp}^\top \boldsymbol{\Lambda}^\top + \mathbf{I}_m)^{-1} \boldsymbol{\Lambda}$        ▷ Representer matrix of $\hat{\mathsf{k}}(x, y)$
    **return** $\hat{\mathsf{k}}(x, y) = \sum_{i,j=1}^m \boldsymbol{\Omega}_{ij} k_\mathcal{H}(z_i, x) k_\mathcal{H}(z_j, y)$        ▷ Correlation kernel
**end procedure**

---

## 4 Implementation

We propose an algorithm for solving the discrete problem (9) associated to (6). To simplify the discussion and relate it to Mariet and Sra [2015], we define the objective $g(\mathbf{X}) = f_n(V^* \mathbf{B}(\mathbf{X}) V) + \lambda \operatorname{Tr}(\mathbf{B}(\mathbf{X}))$ with the change of variables $\mathbf{B}(\mathbf{X}) = \mathbf{R}^{-1\top} \mathbf{X} \mathbf{R}^{-1}$. Then we can rephrase (9) as

$$\min_{\mathbf{X} \succeq 0} g(\mathbf{X}) = -\frac{1}{s} \sum_{\ell=1}^s \log \det(\mathbf{X}_{\mathcal{C}_\ell \mathcal{C}_\ell}) + \log \det \left( \mathbf{I}_{|\mathcal{I}|} + \frac{1}{n} \mathbf{X}_{\mathcal{I}\mathcal{I}} \right) + \lambda \operatorname{Tr}(\mathbf{X} \mathbf{K}^{-1}), \tag{12}$$

where we recall that $n = |\mathcal{I}|$. Define for convenience $\mathbf{U}_\ell$ as the matrix obtained by selecting the columns of the identity matrix which are indexed by $\mathcal{C}_\ell$, so that, we have in particular $\mathbf{X}_{\mathcal{C}_\ell \mathcal{C}_\ell} = \mathbf{U}_\ell^\top \mathbf{X} \mathbf{U}_\ell$. Similarly, define a sampling matrix $\mathbf{U}_\mathcal{I}$ associated to the subset $\mathcal{I}$. Recall the Cholesky decomposition $\mathbf{K} = \mathbf{R}^\top \mathbf{R}$. To minimize (12), we start at some $\mathbf{X}_0 \succ 0$ and use the following iteration

$$\mathbf{X}_{k+1} = \frac{1}{2\lambda} \mathbf{R}^\top \left( \left( \mathbf{I}_m + 4\lambda \mathbf{R}^{-1\top} p(\mathbf{X}_k) \mathbf{R}^{-1} \right)^{1/2} \mathbf{R} - \mathbf{I}_m \right) \mathbf{R}, \tag{13}$$

where $p(\mathbf{X}) = \mathbf{X} + \mathbf{X}\boldsymbol{\Delta}\mathbf{X}$ and $\boldsymbol{\Delta}(\mathbf{X}) = \frac{1}{s}\sum_{\ell=1}^{s}\mathbf{U}_{\ell}\mathbf{X}_{\mathcal{C}_{\ell}\mathcal{C}_{\ell}}^{-1}\mathbf{U}_{\ell}^{\top} - \mathbf{U}_{\mathcal{I}}(\mathbf{X}_{\mathcal{I}\mathcal{I}} + n\mathbf{I}_{|\mathcal{I}|})^{-1}\mathbf{U}_{\mathcal{I}}^{\top}$. We dub this sequence a regularized Picard iteration, as it is a generalization of the Picard iteration which was introduced by Mariet and Sra [2015] in the context of learning discrete L-ensemble DPPs. In Mariet and Sra [2015], the Picard iteration, defined as $\mathbf{X}_{k+1} = p(\mathbf{X}_k)$, is shown to be appropriate for minimizing a different objective given by: $-\frac{1}{s}\sum_{\ell=1}^{s}\log\det(\mathbf{X}_{\mathcal{C}_{\ell}\mathcal{C}_{\ell}}) + \log\det(\mathbf{I} + \mathbf{X})$. The following theorem indicates that the iteration (13) is a good candidate for minimizing $g(\mathbf{X})$.

**Theorem 3.** *Let $\mathbf{X}_k$ for integer $k$ be the sequence generated by (13) and initialized with $\mathbf{X}_0 \succ 0$. Then, the sequence $g(\mathbf{X}_k)$ is monotonically decreasing.*

For a proof, we refer to Section S3.4. In practice, we use the iteration (13) with the inverse change of variables $\mathbf{X}(\mathbf{B}) = \mathbf{R}^{\top}\mathbf{B}\mathbf{R}$ and solve

$$\mathbf{B}_{k+1} = \frac{1}{2\lambda}\left(\left(\mathbf{I}_m + 4\lambda q(\mathbf{B}_k)\right)^{1/2} - \mathbf{I}_m\right), \text{ with } q(\mathbf{B}) = \mathbf{B} + \mathbf{B}\mathbf{R}\boldsymbol{\Delta}\left(\mathbf{X}(\mathbf{B})\right)\mathbf{R}^{\top}\mathbf{B}, \qquad (14)$$

where $\boldsymbol{\Delta}(\mathbf{X})$ is given hereabove. For the stopping criterion, we monitor the objective values of (9) and stop if the relative variation of two consecutive objectives is less than a predefined precision threshold tol. Contrary to (12), the objective (9) does not include the inverse of $\mathbf{K}$, which might be ill-conditioned. The interplay between $\lambda$ and $n$ is best understood by considering (9) with the change of variables $\mathbf{B}' = \mathbf{B}/n$, yielding the equivalent problem

$$\min_{\mathbf{B}' \succeq 0} -\frac{1}{s}\sum_{\ell=1}^{s}\log\det\left(\boldsymbol{\Phi}^{\top}\mathbf{B}'\boldsymbol{\Phi}\right)_{\mathcal{C}_{\ell}\mathcal{C}_{\ell}} + \log\det\left(\mathbf{I} + \boldsymbol{\Phi}^{\top}\mathbf{B}'\boldsymbol{\Phi}\right)_{\mathcal{I}\mathcal{I}} + \lambda n\,\mathrm{Tr}(\mathbf{B}'),$$

where $\boldsymbol{\Phi} = \mathbf{R}$ is a matrix whose $i$-th column is $\boldsymbol{\Phi}_i$ for $1 \leq i \leq m$ as defined in (7). Notice that, up to a $1/n$ factor, $\boldsymbol{\Phi}^{\top}\mathbf{B}'\boldsymbol{\Phi}$ is the in-sample Gram matrix of $\hat{\mathsf{a}}(x, y)$ evaluated on the data set $\mathcal{Z}$; see Algorithm 1. Thus, in the limit $\lambda \to 0$, the above expression corresponds to the MLE estimation of a finite DPP if $\cup_{\ell}\mathcal{C}_{\ell} \subseteq \mathcal{I}$. This is the intuitive connection with finite DPP: the continuous DPP is well approximated by a finite DPP if the ground set $\mathcal{I}$ is a dense enough sampling within $\mathcal{X}$.

## 5 Theoretical guarantees

We now describe the guarantees coming with the approximations presented in the previous section.

**Statistical guarantees for approximating the maximum likelihood problem** Next, we give a statistical guarantee for the approximation of the log-likelihood by its sample version.

**Theorem 4** (Discrete optimal objective approximates full MLE objective). *Let $\mathbf{B}_{\star}$ be the solution of (9). Let $A_{\star}$ be the solution of (5). Let $\delta \in (0, 1/2)$. If $\lambda \geq 2c_n(\delta)$, then with probability at least $1 - 2\delta$, it holds that*

$$|f(A_{\star}) - f_n(V^{*}\mathbf{B}_{\star}V)| \leq \frac{3}{2}\lambda\,\mathrm{Tr}(A_{\star}),$$

*with $0 < c_n \lesssim 1/\sqrt{n}$ given in Theorem 1.*

The above result, proved in Section S3.3, shows that, with high probability, the optimal objective value of the discrete problem is not far from the optimal log-likelihood provided $n$ is large enough. As a simple consequence, the discrete solution also yields a finite rank operator $V\mathbf{B}_{\star}V^{*}$ whose likelihood is not far from the optimal likelihood $f(A_{\star})$, as it can be shown by using a triangle inequality.

**Corollary 5** (Approximation of the full MLE optimizer by a finite rank operator). *Under the assumptions of Theorem 4, if $\lambda \geq 2c_n(\delta)$, with probability at least $1 - 2\delta$, it holds*

$$|f(A_{\star}) - f(V^{*}\mathbf{B}_{\star}V)| \leq 3\lambda\,\mathrm{Tr}(A_{\star})$$

*with $c_n \lesssim 1/\sqrt{n}$ given in Theorem 1.*

The proof of Corollary 5 is also provided in Section S3.3.

**Approximation of the correlation kernel** An important quantity for the control of the amount of points necessary to approximate well the correlation kernel is the so-called *effective dimension* $d_{\mathrm{eff}}(\gamma) = \mathrm{Tr}\left(\mathsf{A}(\mathsf{A} + \gamma\mathbb{I})^{-1}\right)$, which is the expected sample size under the DPP with correlation kernel $\mathsf{K} = \mathsf{A}(\mathsf{A} + \gamma\mathbb{I})^{-1}$.

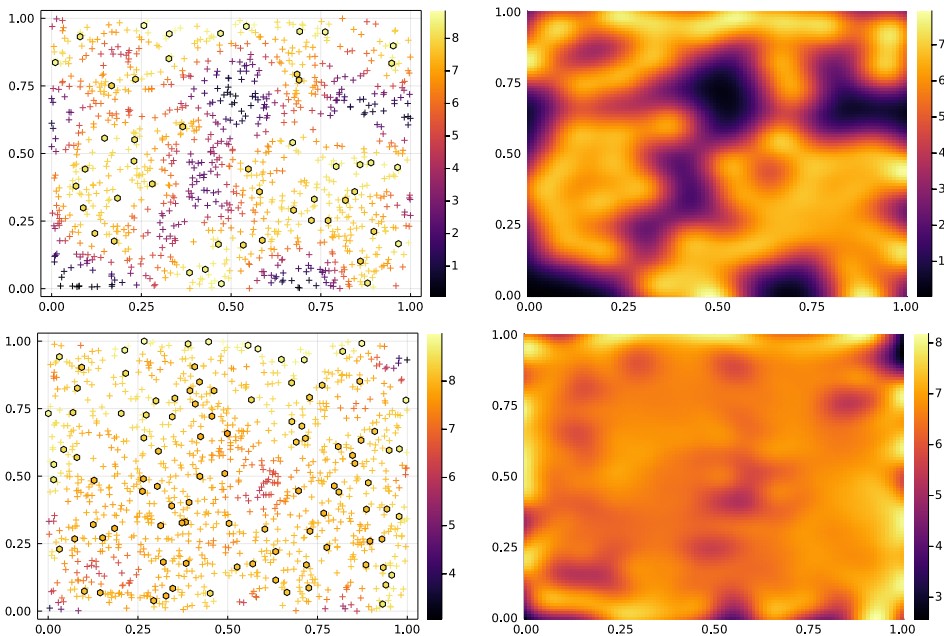

Figure 1: Intensity estimation with $\sigma = 0.1$ and $\lambda = 0.1$ from 1 DPP sample with $\rho = 50$ (top row) and $\rho = 100$ (bottom row). On the LHS, a DPP sample (hexagons) and $n = 1000$ uniform samples (crosses), the color is the diagonal of $\Phi^\top B \Phi$ (in-sample likelihood kernel). On the RHS, out-of-sample estimated intensity $\hat{k}(x, x)$ of the learned process on a $100 \times 100$ grid.

**Theorem 6** (Correlation kernel approximation). *Let $\delta \in (0, 1)$ be a failure probability, let $\epsilon \in (0, 1)$ be an acurracy parameter and let $\gamma > 0$ be a scale factor. Let $\mathsf{K}(\gamma)$ be the correlation kernel (10) defined with $\mathsf{A} = SAS^*$. Consider $\hat{\mathsf{K}}(\gamma)$ defined in (11) with i.i.d. sampling of $p$ points in $\mathcal{X}$ wrt $\mu$. If we take $p \geq \frac{8\kappa^2 \|A\|_{op}}{\gamma\epsilon^2} \log\left(\frac{4 d_{\mathrm{eff}}(\gamma)}{\delta \|\mathsf{K}\|_{op}}\right)$, then, with probability $1 - \delta$, the following multiplicative error bound holds $\frac{1}{1+\epsilon} \mathsf{K}(\gamma) \preceq \hat{\mathsf{K}}(\gamma) \preceq \frac{1}{1-\epsilon} \mathsf{K}(\gamma)$.*

The proof of Theorem 6, given in Section S3.5, mainly relies on a matrix Bernstein inequality. Let us make a few comments. First, we can simply take $\gamma = 1$ in Theorem 6 to recover the common definition of the correlation kernel (1). Second, the presence of $d_{\mathrm{eff}}(\gamma)$ in the logarithm is welcome since it is the expected subset size of the L-ensemble. Third, the quantity $\|A\|_{op}$ directly influences the typical sample size to get an accurate approximation. A worst case bound is $\|A\|_{op} \leq \lambda_{\max}(\mathbf{C})\lambda_{\max}(\mathbf{K})$ with $\mathbf{K} = [k_{\mathcal{H}}(z_i, z_j)]_{1 \leq i,j \leq m}$ and where we used that $A = mS_m \mathbf{C} S_m^*$ in the light of (8). Thus, the lower bound on $p$ may be large in practice. Although probably more costly, an approach inspired from approximate ridge leverage score sampling [Rudi et al., 2018] is likely to allow lower $p$'s. We leave this to future work.

## 6 Empirical evaluation

We consider an L-ensemble with correlation kernel $\mathsf{k}(x, y) = \rho \exp(-\|x - y\|_2^2/\alpha^2)$ defined on $\mathbb{R}^d$ with $\alpha = 0.05$. Following Lavancier et al. [2014], this is a valid kernel if $\rho < (\sqrt{\pi}\alpha)^{-d}$. Note that the intensity, defined as $x \mapsto \mathsf{k}(x, x)$, is constant equal to $\rho$; we shall check that the fitted kernel recovers that property. We draw samples[3] from this continuous DPP in the window $\mathcal{X} = [0, 1]^2$. Two such samples are shown as hexagons in the first column of Figure 1, with respective intensity $\rho = 50$ and $\rho = 100$. For the estimation, we use a Gaussian kernel $k_{\mathcal{H}}(x, y) = \exp\left(-\|x - y\|_2^2/(2\sigma^2)\right)$ with $\sigma > 0$. The computation of the correlation kernel always uses $p = 1000$ uniform samples. Iteration (14) is run until the precision threshold $\mathsf{tol} = 10^{-5}$ is achieved. For stability, we add $10^{-10}$

---

[3]We used the code of Poinas and Lavancier [2021], available at `https://github.com/APoinas/MLEDPP`. It relies on the R package *spatstat* [Baddeley et al., 2015], available under the GPL-2 / GPL-3 licence.

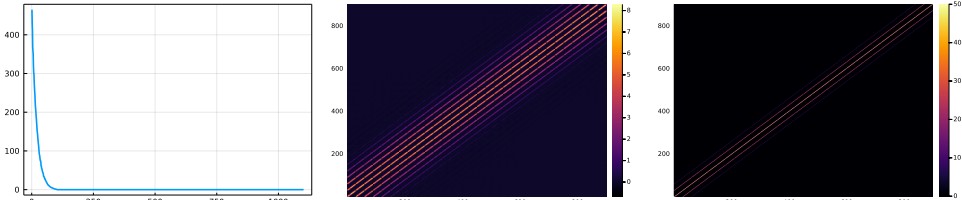

Figure 2: Analysis of the solution corresponding to the example of Figure 1 with $\rho = 100$. Left: eigenvalues of $\mathbf{\Phi}^\top \mathbf{B}\mathbf{\Phi}$. Middle: Gram matrix of $\hat{\mathsf{k}}(x, y)$ on a regular $30 \times 30$ grid within $[0, 1]^2$. Right: Gram matrix of $\mathsf{k}(x, y)$ on the same grid.

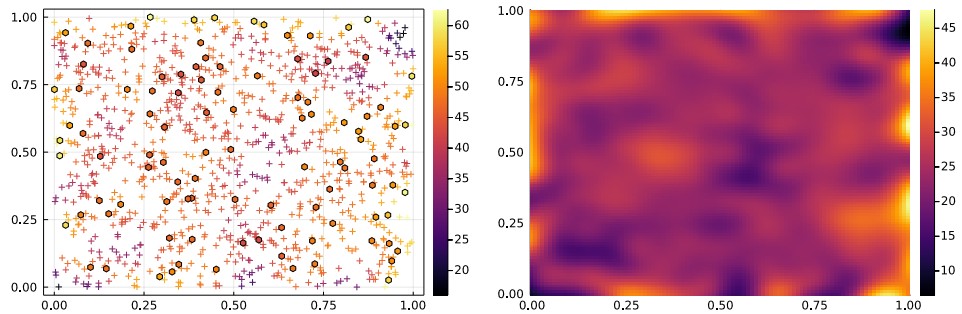

Figure 3: Intensity estimation with $\sigma = 0.1$ and $\lambda = 0.01$ from 1 DPP sample with $\rho = 100$. On the LHS, a DPP sample (hexagons) and $n = 1000$ uniform samples (crosses), the color is the diagonal of $\mathbf{\Phi}^\top \mathbf{B}\mathbf{\Phi}$ (in-sample likelihood kernel). On the RHS, out-of-sample estimated intensity $\hat{\mathsf{k}}(x, x)$ of the learned process on a $100 \times 100$ grid.

to the diagonal of the Gram matrix $\mathbf{K}$. The remaining parameter values are given in captions. We empirically observe that the regularized Picard iteration returns a matrix $\mathbf{B}_\star$ such that $\mathbf{\Phi}^\top \mathbf{B}_\star \mathbf{\Phi}$ is low rank; see Figure 2 (left). A lesson from Figure 1 is that the sample size of the DPP has to be large enough to retrieve a constant intensity $\hat{k}(x, x)$. In particular, the top row of this figure illustrates a case where $\sigma$ is too small. Also, due to the large regularization $\lambda = 0.1$ and the use of only one DPP sample, the scale of $\rho$ is clearly underestimated in this example. On the contrary, in Figure 3, for a smaller regularization parameter $\lambda = 0.01$, the intensity scale estimate is larger. We also observe that a large regularization parameter tends to smooth out the local variations of the intensity, which is not surprising. A comparison between a Gram matrix of $\hat{\mathsf{k}}(x, y)$ and $\mathsf{k}(x, y)$ is given in Figure 2 corresponding to the example of Figure 1. The short-range diagonal structure is recovered, while some long-range structures are smoothed out. More illustrative simulations are given in Section S4, with a study of the influence of the hyperparameters, including the use of $s > 1$ DPP samples. In particular, the estimation of the intensity is improved if several DPP samples are used with a smaller value of $\lambda$.

## 7    Discussion

We leveraged recent progress on kernel methods to propose a nonparametric approach to learning continuous DPPs. We see three major limitations of our procedure. First, our final objective function is nonconvex, and our algorithm is only guaranteed to increase its objective function. Experimental evidence suggests that our approach recovers the synthetic kernel, but more work is needed to study the maximizers of the likelihood, in the spirit of Brunel et al. [2017a] for finite DPPs, and the properties of our fixed point algorithm. Second, the estimated integral kernel does not have any explicit structure, other than being implicitly forced to be low-rank because of the trace penalty. Adding structural assumptions might be desirable, either for modelling or learning purposes. For modelling, it is not uncommon to assume that the underlying continuous DPP is stationary, for example, which implies that the correlation kernel $\mathsf{k}(x, y)$ depends only on $x - y$. For learning, structural assumptions on the kernel may reduce the computational cost, or reduce the number of

maximizers of the likelihood. The third limitation of our pipeline is that, like most nonparametric methods, it still requires to tune a handful of hyperparameters, and, in our experience, the final performance varies significantly with the lengthscale of the RKHS kernel or the coefficient of the trace penalty. An automatic tuning procedure with guarantees would make the pipeline turn-key.

Maybe unexpectedly, future work could also deal with transferring our pipeline to the finite DPP setting. Indeed, we saw in Section 4 that in some asymptotic regime, our regularized MLE objective is close to a regularized version of the MLE objective for a finite DPP. Investigating this maximum a posteriori inference problem may shed some new light on nonparametric inference for finite DPPs. Intuitively, regularization should improve learning and prediction when data is scarce.

## Acknowledgements

We thank Arnaud Poinas and Guillaume Gautier for useful discussions on the manuscript. We acknowledge support from ERC grant BLACKJACK (ERC-2019-STG-851866) and ANR AI chair BACCARAT (ANR-20-CHIA-0002).

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
