# Supplementary material to *Nonparametric estimation of continuous DPPs with kernel methods*

**Michaël Fanuel and Rémi Bardenet**
Université de Lille, CNRS, Centrale Lille
UMR 9189 – CRIStAL, F-59000 Lille, France
{michael.fanuel, remi.bardenet}@univ-lille.fr

For ease of reference, sections, propositions and equations that belong to this supplementary material are prefixed with an 'S'. Additionally, labels in light blue refer to the main paper. Hyperlinks across documents should work if the two PDFs are placed in the same folder.

**Roadmap.** In Section S1, we present useful technical results. Next, in a specific case, we show that the discrete problem (6) admits a closed-form solution that we discuss in Section S2. Noticeably, this special case allows the understanding of the behaviour of the estimated DPP kernel in both the small and large regularization ($\lambda$) limits. In Section S3, all the deferred proofs are given. Finally, Section S4 provides a finer analysis of the empirical results of Section 6 as well as a description of the convergence of the regularized Picard algorithm to the closed-form solution described in Section S2.

## S1 Useful technical results

### S1.1 Technical lemmata

We preface the proofs of our main results with three lemmata.

**Lemma S1.** *Let $k$ be a strictly positive definite kernel of a RKHS $\mathcal{H}$ and let $\mathbf{C}$ be a $n \times n$ symmetric matrix. Assume that $\{x_i\}_{1 \leq i \leq n}$ are such that the Gram matrix $\mathbf{K} = [k_{\mathcal{H}}(x_i, x_j)]_{1 \leq i, j \leq n}$ is non-singular. Then, the non-zero eigenvalues of $\sum_{i,j=1}^n \mathbf{C}_{ij} \phi(x_i) \otimes \overline{\phi(x_j)}$ correspond to the non-zero eigenvalues of $\mathbf{KC}$.*

*Proof.* By definition of the sampling operator (see Section 1.1), it holds that $S_n^* \mathbf{C} S_n = \frac{1}{n} \sum_{i,j=1}^n \mathbf{C}_{ij} \phi(x_i) \otimes \overline{\phi(x_j)}$ and we have $S_n S_n^* \mathbf{C} = \frac{1}{n} \mathbf{KC}$. Thus, we need to show that the non-zero eigenvalues $S_n^* \mathbf{C} S_n$ correspond to the non-zero eigenvalues of $S_n S_n^* \mathbf{C}$.

Let $g_\lambda \in \mathcal{H}$ be an eigenvector of $S_n^* \mathbf{C} S_n$ with eigenvalue $\lambda \neq 0$. First, we show that $S_n g_\lambda$ is an eigenvector of $S_n S_n^* \mathbf{C}$ with eigenvalue $\lambda$. We have $S_n^* \mathbf{C} S_n g_\lambda = \lambda g_\lambda$. By acting on both sides of the latter equation with $S_n$, we find $S_n (S_n^* \mathbf{C} S_n) g_\lambda = \lambda S_n g_\lambda$. This is equivalent to $S_n S_n^* \mathbf{C} (S_n g_\lambda) = \lambda (S_n g_\lambda)$. Second, since $S_n S_n^* \succ 0$, remark that $S_n S_n^* \mathbf{C}$ is related by a similarity to $(S_n S_n^*)^{1/2} \mathbf{C} (S_n S_n^*)^{1/2}$, which is diagonalizable. Since $S_n^* \mathbf{C} S_n$ is at most of rank $n$, the non-zero eigenvalues of $S_n^* \mathbf{C} S_n$ match the non-zero eigenvalues of $(S_n S_n^*)^{1/2} \mathbf{C} (S_n S_n^*)^{1/2}$, which in turn are the same as the non-zero eigenvalues of $S_n S_n^* \mathbf{C}$. □

**Lemma S2.** *Let $\mathbf{\Sigma} \in \mathcal{S}_+(\mathbb{R}^m)$ and let $\mathcal{I}$ be a subset of $\{1, \dots, m\}$. Then, the function*

$$\mathbf{\Sigma} \mapsto \log \det(\mathbf{\Sigma}) + \log \det(\mathbf{I}_m + \mathbf{\Sigma}^{-1}/|\mathcal{I}|)_{\mathcal{I}\mathcal{I}} \tag{S1}$$

*is strictly concave on $\{\mathbf{\Sigma} \succ 0\}$.*

*Proof.* To simplify the expression, we do the change of variables $\mathbf{\Sigma} \mapsto \mathbf{\Sigma}/|\mathcal{I}|$ and analyse $\log \det(\mathbf{\Sigma}) + \log \det(\mathbf{I}_m + \mathbf{\Sigma}^{-1})_{\mathcal{I}\mathcal{I}}$ which differs from the original function by an additive constant.

Let $\mathbf{U}_{\mathcal{I}}$ be the matrix obtained by selecting the columns of the identity matrix which are indexed by $\mathcal{I}$. We rewrite the second term in (S1) as

$$\log\det(\mathbf{I}_m + \mathbf{\Sigma}^{-1})_{\mathcal{II}} = \log\det(\mathbf{I}_{|\mathcal{I}|} + \mathbf{U}_{\mathcal{I}}^\top \mathbf{\Sigma}^{-1} \mathbf{U}_{\mathcal{I}}) = \log\det(\mathbf{I}_m + \mathbf{\Sigma}^{-1/2}\mathbf{U}_{\mathcal{I}}\mathbf{U}_{\mathcal{I}}^\top\mathbf{\Sigma}^{-1/2}),$$

by Sylvester's identity. This leads to

$$\log\det(\mathbf{\Sigma}) + \log\det(\mathbf{I}_m + \mathbf{\Sigma}^{-1})_{\mathcal{II}} = \log\det(\mathbf{\Sigma} + \mathbf{U}_{\mathcal{I}}\mathbf{U}_{\mathcal{I}}^\top).$$

To check that $\log\det(\mathbf{\Sigma} + \mathbf{U}_{\mathcal{I}}\mathbf{U}_{\mathcal{I}}^\top)$ is strictly concave on $\{\mathbf{\Sigma} \succ 0\}$, we verify that its Hessian is negative any direction $\mathbf{H}$. Its first order directional derivative reads $\mathrm{Tr}\left((\mathbf{\Sigma} + \mathbf{U}_{\mathcal{I}}\mathbf{U}_{\mathcal{I}}^\top)^{-1}\mathbf{H}\right)$. Hence, the second order directional derivative in the direction of $\mathbf{H}$ writes

$$-\mathrm{Tr}\left((\mathbf{\Sigma} + \mathbf{U}_{\mathcal{I}}\mathbf{U}_{\mathcal{I}}^\top)^{-1}\mathbf{H}(\mathbf{\Sigma} + \mathbf{U}_{\mathcal{I}}\mathbf{U}_{\mathcal{I}}^\top)^{-1}\mathbf{H}\right),$$

which is indeed a negative number. $\qquad\square$

The following result is borrowed from Mariet and Sra [2015, Lemma 2.3.].

**Lemma S3.** *Let $\mathbf{\Sigma} \in \mathcal{S}_+(\mathbb{R}^m)$ and let $\mathbf{U} \in \mathbb{R}^{m\times\ell}$ be a matrix with $\ell \leq m$ orthonormal columns. Then, the function $-\log\det(\mathbf{U}^\top\mathbf{\Sigma}^{-1}\mathbf{U})$ is concave on $\mathbf{\Sigma} \succ 0$.*

*Proof.* The function $-\log\det(\mathbf{U}^\top\mathbf{\Sigma}^{-1}\mathbf{U})$ is concave on $\mathbf{\Sigma} \succ 0$ since $\log\det(\mathbf{U}^\top\mathbf{\Sigma}^{-1}\mathbf{U})$ is convex on $\mathbf{\Sigma} \succ 0$ for any $\mathbf{U}$ such that $\mathbf{U}^\top\mathbf{U} = \mathbf{I}$ as stated in Mariet and Sra [2015, Lemma 2.3.]. $\qquad\square$

## S1.2 Use of the representer theorem

We here clarify the definition of the representer theorem used in this paper.

### S1.2.1 Extended representer theorem

In Section 3, we used a slight extension of the representer theorem of Marteau-Ferey et al. [2020] which we clarify here. Let us first define some notations. Let $\mathcal{H}$ be a RKHS with feature map $\phi(\cdot)$ and $\{z_1, \ldots, z_m\}$ be a data set such as defined in Section 1.1. Define

$$h_A(z) = \langle\phi(z), A\phi(z)\rangle \text{ and } h_A(z, z') = \langle\phi(z), A\phi(z')\rangle.$$

In this paper, we consider the problem

$$\min_{A\in\mathcal{S}_+(\mathcal{H})} L\left(h_A(z_i, z_j)\right)_{1\leq i,j\leq m} + \mathrm{Tr}(A), \tag{S2}$$

where $L$ is a loss function (specified below). In contrast, the first term of the problem considered in Marteau-Ferey et al. [2020] is of the following form: $L(h_A(z_i))_{1\leq i\leq m}$. In other words, the latter loss function involves only diagonal elements $\langle\phi(z_i), A\phi(z_i)\rangle$ for $1 \leq i \leq m$ while (S2) also involves off-diagonal elements. Now, denote by $\Pi_m$ the projector on $\mathrm{span}\{\phi(z_i), i = 1, \ldots, m\}$, and define

$$\mathcal{S}_{m,+}(\mathcal{H}) = \{\Pi_m A\Pi_m : A \in \mathcal{S}_+(\mathcal{H})\}.$$

Then, we have the following proposition.

**Proposition S4** (Extension of Proposition 7 in Marteau-Ferey et al. [2020])**.** *Let $L$ be a lower semi-continuous function such that $L(h_A(z_i, z_j))_{1\leq i,j\leq m} + \mathrm{Tr}(A)$ is bounded below. Then (S2) has a solution $A_\star$ which is in $\mathcal{S}_{m,+}(\mathcal{H})$.*

*Proof sketch.* The key step is the following identity

$$h_A(z_i, z_j) = \langle\phi(z_i), A\phi(z_j)\rangle = \langle\phi(z_i), \Pi_m A\Pi_m\phi(z_j)\rangle = h_{\Pi_m A\Pi_m}(z_i, z_j), \ (1 \leq i, j \leq m)$$

which is a direct consequence of the definition of $\Pi_m$. Also, we have $\mathrm{Tr}(\Pi_m A\Pi_m) \leq \mathrm{Tr}(A)$. The remainder of the proof follows exactly the same lines as in Marteau-Ferey et al. [2020]. Notice that, compared with Marteau-Ferey et al. [2020, Proposition 7], we do not require the loss $L$ to be lower bounded but rather ask the full objective to be lower bounded, which is a weaker assumption but does not alter the argument. $\qquad\square$

### S1.2.2 Applying the extended representer theorem

Now, let us prove that the objective of (6), given by $f_n(A) + \lambda \operatorname{Tr}(A)$, is lower bounded. We recall that

$$f_n(A) = -\frac{1}{s} \sum_{\ell=1}^{s} \log \det \left[ \langle \phi(x_i), A\phi(x_j) \rangle \right]_{i,j \in \mathcal{C}_\ell} + \log \det(\mathbf{I}_n + S_n A S_n^*).$$

For clarity, we recall that $\mathcal{C} \triangleq \cup_{\ell=1}^{s} \mathcal{C}_\ell$ and $\mathcal{I} = \{x_1', \ldots, x_n'\}$. Then, write the set of points $\mathcal{Z} \triangleq \mathcal{C} \cup \mathcal{I}$ as $\{z_1, \ldots, z_m\}$ with $m = |\mathcal{C}| + n$. Recall that we placed ourselves on an event happening with probability one where the sets $\mathcal{C}_1, \ldots, \mathcal{C}_s, \mathcal{I}$ are disjoint. Now, we denote by $\Pi_m$ the orthogonal projector on $\operatorname{span}\{\phi(z_i) : 1 \leq i \leq m\}$, which writes

$$\Pi_m = \sum_{i,j=1}^{m} (K^{-1})_{ij} \phi(z_i) \otimes \overline{\phi(z_j)}. \tag{S3}$$

First, we have $\log \det(\mathbf{I}_n + S_n A S_n^*) \geq 0$. Second, we use

$$\operatorname{Tr}(A) = \operatorname{Tr}\left(\Pi_m A \Pi_m\right) + \operatorname{Tr}\left(\Pi_m^{\perp} A \Pi_m^{\perp}\right) \geq \operatorname{Tr}\left(\Pi_m A \Pi_m\right) = \operatorname{Tr}\left(\Pi_m A\right).$$

Thus, a lower bound on the objective (6) is

$$f_n(A) + \lambda \operatorname{Tr}(A) \geq -\frac{1}{s} \sum_{\ell=1}^{s} \log \det \left[ \langle \phi(x_i), A\phi(x_j) \rangle \right]_{i,j \in \mathcal{C}_\ell} + \lambda \operatorname{Tr}\left(\Pi_m A\right).$$

Now we define the matrix $M$ with elements $M_{ij} = \langle \phi(z_i), A\phi(z_j) \rangle$ for $1 \leq i, j \leq m$ and notice that

$$\operatorname{Tr}\left(\Pi_m A\right) = \operatorname{Tr}(M K^{-1}) \geq \operatorname{Tr}(M)/\lambda_{\max}(K).$$

Remark that the matrix defined by $M^{(\ell)} = \left[ \langle \phi(x_i), A\phi(x_j) \rangle \right]_{i,j \in \mathcal{C}_\ell}$ associated with the $\ell$-th DPP sample for $1 \leq \ell \leq s$ is a *principal* submatrix of the $m \times m$ matrix $M = \left[ \langle \phi(z_i), A\phi(z_j) \rangle \right]_{1 \leq i,j \leq m}$. Since the sets $\mathcal{C}_\ell$ are disjoint, we have $\sum_{\ell=1}^{s} \operatorname{Tr}(M^{(\ell)}) \leq \operatorname{Tr}(M)$. Denoting $\lambda' = \frac{\lambda}{\lambda_{\max}(K)}$, by using the last inequality, we find

$$f_n(A) + \lambda \operatorname{Tr}(A) \geq -\frac{1}{s} \sum_{\ell=1}^{s} \log \det M^{(\ell)} + \lambda' \sum_{\ell=1}^{s} \operatorname{Tr}(M^{(\ell)})$$

$$= \frac{1}{s} \sum_{\ell=1}^{s} \left( -\log \det M^{(\ell)} + s\lambda' \operatorname{Tr}(M^{(\ell)}) \right).$$

Finally, we use the following proposition to show that each term in the above sum is bounded from below.

**Proposition S5.** *Let $a > 0$. The function $h(\sigma) = -\log(\sigma) + a\sigma$ satisfies $h(\sigma) \geq h(1/a) = 1 + \log(a)$ for all $\sigma > 0$.*

Thus a lower bound for the objective of (6) is obtained by applying Proposition S5 with $a = s\lambda/\lambda_{\max}(K)$ to the each eigenvalue of $M^{(\ell)}$ for all $1 \leq \ell \leq s$.

### S1.3 Boundedness of the discrete objective function

At first sight, we may wonder if the objective of the optimization problem (12) is lower bounded. We show here that the optimization problem is well-defined for all regularization parameters $\lambda > 0$. The lower boundedness of the discrete objective follows directly from Section S1.2.2 in the case of a finite rank operator. For completeness, we repeat below the argument in the discrete case. Explicitly, this discrete objective reads

$$g(\mathbf{X}) = -\frac{1}{s} \sum_{\ell=1}^{s} \log \det(\mathbf{X}_{\mathcal{C}_\ell \mathcal{C}_\ell}) + \log \det \left( \mathbf{I}_{|\mathcal{I}|} + \frac{1}{n} \mathbf{X}_{\mathcal{I}\mathcal{I}} \right) + \lambda \operatorname{Tr}(\mathbf{X} K^{-1}),$$

for all $\mathbf{X} \succeq 0$. First, since $\mathbf{X}_{\mathcal{II}} \succeq 0$, we have $\log \det \left( \mathbf{I}_{|\mathcal{I}|} + \frac{1}{n} \mathbf{X}_{\mathcal{II}} \right) \geq 0$. Next, we use that $\mathbf{K} \succ 0$ by assumption, which implies that $\mathrm{Tr}(\mathbf{X}\mathbf{K}^{-1}) \geq \mathrm{Tr}(\mathbf{X})/\lambda_{\max}(\mathbf{K})$. Denote $\lambda' = \frac{\lambda}{\lambda_{\max}(\mathbf{K})}$. Thus, we can lower bound the objective function as follows:

$$g(\mathbf{X}) \geq -\frac{1}{s} \sum_{\ell=1}^{s} \log \det(\mathbf{X}_{\mathcal{C}_\ell \mathcal{C}_\ell}) + \lambda' \, \mathrm{Tr}(\mathbf{X})$$

$$\geq \frac{1}{s} \sum_{\ell=1}^{s} \big( -\log \det(\mathbf{X}_{\mathcal{C}_\ell \mathcal{C}_\ell}) + \lambda' s \, \mathrm{Tr}(\mathbf{X}_{\mathcal{C}_\ell \mathcal{C}_\ell}) \big),$$

where we used that $\mathrm{Tr}(\mathbf{X}) = \mathrm{Tr}(\mathbf{X}_{\mathcal{II}}) + \sum_{\ell=1}^{s} \mathrm{Tr}(\mathbf{X}_{\mathcal{C}_\ell \mathcal{C}_\ell})$ with $\mathrm{Tr}(\mathbf{X}_{\mathcal{II}}) \geq 0$ for $\mathbf{X} \succeq 0$. Hence, by using Proposition S5, $g(\mathbf{X})$ is bounded from below by a sum of functions which are lower bounded.

## S2 Extra result: an explicit solution for the single-sample case

We note that if the dataset is made of only one sample (i.e., $s = 1$), and if that sample is also used to approximate the Fredholm determinant (i.e., $\mathcal{I} = \mathcal{C}$), then the problem (6) admits an explicit solution.

**Proposition S6.** *Let $\mathcal{C} = \{x_i \in \mathcal{X}\}_{1 \leq i \leq m}$ be such that the kernel matrix $\mathbf{K} = [k_{\mathcal{H}}(x_i, x_j)]_{1 \leq i,j \leq m}$ is invertible. Consider the problem (6) with $\mathcal{I} = \mathcal{C}$ and $\lambda > 0$. Then the solution of (6) has the form $A_\star = \sum_{i,j=1}^{m} \mathbf{C}_{\star ij} \phi(x_i) \otimes \overline{\phi(x_j)}$, with*

$$\mathbf{C}_\star = \mathbf{C}_\star(\lambda) = \frac{1}{2} \mathbf{K}^{-2} \left( (m^2 \mathbf{I}_m + 4m\mathbf{K}/\lambda)^{1/2} - m\mathbf{I}_m \right).$$

*Proof.* By the representer theorem in Marteau-Ferey et al. [2020, Theorem 1], the solution is of the form $A = \sum_{i,j=1}^{m} \mathbf{C}_{ij} \phi(x_i) \otimes \overline{\phi(x_j)}$ where $\mathbf{C}$ is the solution of

$$\min_{\mathbf{C} \succ 0} -\log \det (\mathbf{K}\mathbf{C}\mathbf{K}) + \log \det (\mathbf{I}_m + \mathbf{K}\mathbf{C}\mathbf{K}/m) + \lambda \, \mathrm{Tr}(\mathbf{K}\mathbf{C}),$$

Define $\mathbf{X} = \mathbf{K}\mathbf{C}\mathbf{K}$ and denote the objective by $g(\mathbf{X}) = -\log \det(\mathbf{X}) + \log \det(\mathbf{I}_m + \frac{1}{m}\mathbf{X}) + \lambda \, \mathrm{Tr}(\mathbf{K}^{-1}\mathbf{X})$. Let $\mathbf{H}$ be a $m \times m$ symmetric matrix. By taking a directional derivative of the objective in the direction $\mathbf{H}$, we find a first order condition $\mathrm{Tr}\left[ D_{\mathbf{H}} g(\mathbf{X}) \mathbf{H} \right] = 0$, with $D_{\mathbf{H}} g(\mathbf{X}) = -\mathbf{X}^{-1} + (\mathbf{X} + m\mathbf{I}_m)^{-1} + \lambda \mathbf{K}^{-1}$. Since this condition should be satisfied for all $\mathbf{H}$, we simply solve the equation $D_{\mathbf{H}} g(\mathbf{X}) = 0$. Thanks to the invertibility of $\mathbf{K}$, we have equivalently

$$\mathbf{X}^2 + m\mathbf{X} - \frac{m}{\lambda}\mathbf{K} = 0.$$

A simple algebraic manipulation yields $\mathbf{X}_\star = \frac{1}{2}\left((m^2 \mathbf{I}_m + 4m\mathbf{K}/\lambda)^{1/2} - m\mathbf{I}_m\right) \succ 0$. To check that $\mathbf{X}_\star$ is minimum, it is sufficient to analyse the Hessian of the objective in any direction $\mathbf{H}$. Let $t \geq 0$ small enough such that $\mathbf{X} + t\mathbf{H} \succ 0$. The second order derivative of $g(\mathbf{X} + t\mathbf{H})$ writes

$$\frac{\mathrm{d}^2}{\mathrm{d}t^2} g(\mathbf{X} + t\mathbf{H})|_{t=0} = \mathrm{Tr}(\mathbf{H}\mathbf{X}^{-1}\mathbf{H}\mathbf{X}^{-1}) - \mathrm{Tr}(\mathbf{H}(\mathbf{X} + m\mathbf{I}_m)^{-1}\mathbf{H}(\mathbf{X} + m\mathbf{I}_m)^{-1}).$$

By using the first order condition $\mathbf{X}_\star^{-1} = (\mathbf{X}_\star + m\mathbf{I}_m)^{-1} + \lambda \mathbf{K}^{-1}$, we find

$$\frac{\mathrm{d}^2}{\mathrm{d}t^2} g(\mathbf{X}_\star + t\mathbf{H})|_{t=0} = \mathrm{Tr}\left( \mathbf{H}(\lambda \mathbf{K}^{-1})\mathbf{H}\big(\lambda \mathbf{K}^{-1} + 2(\mathbf{X}_\star + m\mathbf{I}_m)^{-1}\big) \right)$$

$$= \mathrm{Tr}\left( \mathbf{M}\mathbf{N}\mathbf{M} \right).$$

with $\mathbf{M} = (\lambda \mathbf{K}^{-1})^{1/2}\mathbf{H}(\lambda \mathbf{K}^{-1})^{1/2}$ and

$$\mathbf{N} = \mathbf{I} + 2(\lambda \mathbf{K}^{-1})^{-1/2}(\mathbf{X}_\star + m\mathbf{I}_m)^{-1}(\lambda \mathbf{K}^{-1})^{-1/2} \succ 0.$$

Notice that $\mathbf{M}\mathbf{N}\mathbf{M} \succeq 0$ since $\mathbf{N} \succ 0$. Therefore, it holds $\frac{\mathrm{d}^2}{\mathrm{d}t^2} g(\mathbf{X}_\star + t\mathbf{H})|_{t=0} \geq 0$. Since this is true for any direction $\mathbf{H}$, the matrix $\mathbf{X}_\star$ is a local minimum of $g(\mathbf{X})$. $\square$

While the exact solution of Proposition S6 is hard to intepret, if the regularization parameter goes to zero, the estimated correlation kernel tends to that of a projection DPP. In what follows, the notation $f(\lambda) \sim g(\lambda)$ stands for $\lim_{\lambda \to +\infty} f(\lambda)/g(\lambda) = 1$.

**Proposition S7** (Projection DPP in the low regularization limit). *Under the assumptions of Proposition S6, the correlation kernel* $\mathsf{K}(\lambda) = SA_\star(\lambda)S^*(SA_\star(\lambda)S^* + \mathbb{I})^{-1}$, *with* $A_\star(\lambda) = \sum_{i,j=1}^m \mathbf{C}_{\star ij}(\lambda)\phi(x_i) \otimes \overline{\phi(x_j)}$ *and* $C_\star(\lambda)$ *given in Proposition S6, has a range that is independent of* $\lambda$. *Furthermore,* $\mathsf{K}(\lambda)$ *converges to the projection operator onto that range when* $\lambda \to 0$. *In particular, the integral kernel of* $A_\star(\lambda)$ *has the following asymptotic expansion*

$$\mathsf{a}_\star(x,y) \sim (m/\lambda)^{1/2}\mathbf{k}_x^\top \mathbf{K}^{-3/2}\mathbf{k}_y \text{ as } \lambda \to 0,$$

*pointwisely, with* $\mathbf{k}_x = [k_\mathcal{H}(x, x_1) \dots, k_\mathcal{H}(x, x_m)]^\top$.

*Proof.* As shown in Lemma S1, the non-zero eigenvalues of $A_\star(\lambda)$ are the eigenvalues of $\mathbf{K}\mathbf{C}_\star(\lambda)$ since $\mathbf{K}$ is non-singular. The eigenvalues of $\mathbf{K}\mathbf{C}_\star(\lambda)$ are $\sigma_\ell(\lambda) = \frac{m}{2\varsigma_\ell}(\sqrt{1 + 4\frac{\varsigma_\ell}{m\lambda}} - 1)$ where $\varsigma_\ell$ is the $\ell$-th eigenvalue of $\mathbf{K}$ for $1 \le \ell \le m$. Thus, the operator $A_\star(\lambda)$ is of finite rank, i.e., $A_\star(\lambda) = \sum_{\ell=1}^m \sigma_\ell(\lambda)v_\ell \otimes \overline{v_\ell}$, where the normalized eigenvectors $v_\ell$ are independent of $\lambda$. The eigenvalues satisfy $\sigma_\ell(\lambda) \to +\infty$ as $\lambda \to 0$, such that $\lim_{\lambda \to 0} \sigma_\ell(\lambda)\sqrt{\lambda}$ is finite. Note that $A_\star(\lambda) = \sum_{\ell=1}^m \sigma_\ell(\lambda)(Sv_\ell) \otimes \overline{(Sv_\ell)}$ is of finite rank and $Sv_\ell$ does not depend on $\lambda$. Hence, the operator

$$\mathsf{K}(\lambda) = \sqrt{\lambda}SA_\star(\lambda)S^* \left(\sqrt{\lambda}SA_\star(\lambda)S^* + \sqrt{\lambda}\mathbb{I}\right)^{-1},$$

converges to the projector on the range of $SA_\star(\lambda)S^*$ as $\lambda \to 0$. □

On the other hand, as the regularization parameter goes to infinity, the integral kernel of $\mathsf{A}_\star(\lambda)$ behaves asymptotically as an out-of-sample Nyström approximation.

**Proposition S8** (Large regularization limit). *Under the assumptions of Proposition S6, the integral kernel of* $\mathsf{A}_\star(\lambda)$ *has the following asymptotic expansion*

$$\mathsf{a}_\star(x,y) \sim \lambda^{-1}\mathbf{k}_x^\top \mathbf{K}^{-1}\mathbf{k}_y \text{ as } \lambda \to +\infty,$$

*pointwisely, with* $\mathbf{k}_x = [k_\mathcal{H}(x, x_1) \dots, k_\mathcal{H}(x, x_m)]^\top$.

The proof of this result is a simple consequence of the series expansion $\sqrt{1 + 4x} = 1 + 2x + O(x^2)$.

## S3 Deferred proofs

### S3.1 Proof of Lemma 2

*Proof of Lemma 2.* Since $V^*V$ is the orthogonal projector onto the span of $\phi(z_i)$, $1 \le i \le m$, we have
$$\boldsymbol{\Phi}_i^\top \bar{\mathbf{B}} \boldsymbol{\Phi}_j = \langle V^*V\phi(x_i), AV^*V\phi(x_j)\rangle = \langle \phi(x_i), A\phi(x_j)\rangle.$$
By Sylvester's identity, and since $\|V^*V\|_{op} \le 1$, it holds
$$\log\det(\mathbf{I}_m + \bar{\mathbf{B}}) = \log\det(\mathbb{I} + VAV^*) = \log\det(\mathbb{I} + V^*VA) \le \log\det(\mathbb{I} + A).$$
Similarly, it holds that $\text{Tr}(\bar{\mathbf{B}}) = \text{Tr}(V^*VA) \le \text{Tr}(A)$. □

### S3.2 Approximation of the Fredholm determinant

Before giving the proof of Theorem 1, we remind a well-known and useful formula for the Fredholm determinant of an operator. Let $M$ be a trace class endomorphism of a separable Hilbert space. Denote by $\lambda_\ell(M)$ its (possibly infinitely many) eigenvalues for $\ell \ge 1$, including multiplicities. Then, we have
$$\det(\mathbb{I} + M) = \prod_\ell (1 + \lambda_\ell(M)),$$
where the (infinite) product is uniformly convergent, see Bornemann [2010, Eq. (3.1)] and references therein. Now, let $\mathcal{F}_1$ and $\mathcal{F}_2$ be two separable Hilbert spaces. Let $T : \mathcal{F}_1 \to \mathcal{F}_2$ such that $T^*T$ is trace class. Note that $T^*T$ is self-adjoint and positive semi-definite. Then, $T^*T$ and $TT^*$ have the same non-zero eigenvalues (Jacobson's lemma) and Sylvester's identity
$$\det(\mathbb{I} + TT^*) = \det(\mathbb{I} + T^*T),$$
holds. This can be seen as a consequence of the singular value decomposition of a compact operator; see Weidmann [1980, page 170, Theorem 7.6]. We now give the proof of Theorem 1.

*Proof of Theorem 1.* To begin, we recall two useful facts. First, If $M$ and $N$ are two trace class endomorphisms of the same Hilbert space, then $\log\det(\mathbb{I} + M) + \log\det(\mathbb{I} + N) = \log\det((\mathbb{I} + M)(\mathbb{I} + N))$; see e.g. Simon [1977, Thm 3.8]. Second, if $M$ is a trace class endomorphisms of a Hilbert space, we have $(\mathbb{I} + M)^{-1} = \mathbb{I} - M(M + \mathbb{I})^{-1}$, and thus, $\det((\mathbb{I} + M)^{-1})$ is a well-defined Fredholm determinant since $M(M + \mathbb{I})^{-1}$ is trace class. Now, define $d = \log\det(\mathbb{I} + SAS^*)$ and $d_n = \log\det(\mathbf{I}_n + S_n A S_n^*)$. Using Sylvester's identity and the aforementioned facts, we write

$$
\begin{aligned}
d_n - d &= \log\det(\mathbf{I}_n + S_n A S_n^*) - \log\det(\mathbb{I} + SAS^*) \\
&= \log\det(\mathbb{I} + A^{1/2} S_n^* S_n A^{1/2}) - \log\det(\mathbb{I} + A^{1/2} S^* S A^{1/2}) \text{ (Sylvester's identity)} \\
&= \log\det\left((\mathbb{I} + A^{1/2} S_n^* S_n A^{1/2})(\mathbb{I} + A^{1/2} S^* S A^{1/2})^{-1}\right).
\end{aligned}
\tag{S4}
$$

By a direct substitution, we verify that $d_n - d = \log\det\left(\mathbb{I} + E_n\right),$ with

$$
E_n = (\mathbb{I} + A^{1/2} S^* S A^{1/2})^{-1/2} A^{1/2} (S_n^* S_n - S^* S) A^{1/2} (\mathbb{I} + A^{1/2} S^* S A^{1/2})^{-1/2}.
$$

Note that, in view of (S4), $\det\left(\mathbb{I} + E_n\right)$ is a positive real number. Next, by using $S_n^* S_n - S^* S \preceq \|S_n^* S_n - S^* S\|_{op}\mathbb{I}$ and Sylvester's identity, we find

$$
\begin{aligned}
d_n - d &\leq \log\det\left(\mathbb{I} + \|S_n^* S_n - S^* S\|_{op}(\mathbb{I} + A^{1/2} S^* S A^{1/2})^{-1/2} A(\mathbb{I} + A^{1/2} S^* S A^{1/2})^{-1/2}\right) \\
&= \log\det\left(\mathbb{I} + \|S_n^* S_n - S^* S\|_{op} A^{1/2}(\mathbb{I} + A^{1/2} S^* S A^{1/2})^{-1} A^{1/2}\right) \\
&\leq \log\det(\mathbb{I} + \|S_n^* S_n - S^* S\|_{op} A).
\end{aligned}
$$

The latter bound is finite, since for $M$ a trace-class operator, we have $|\det(\mathbb{I} + M)| \leq \exp(\|M\|_\star)$, where $\|M\|_\star$ is the trace (or nuclear) norm. By exchanging the roles of $S_n^* S_n$ and $S^* S$, we also find

$$
d - d_n \leq \log\det(\mathbb{I} + \|S_n^* S_n - S^* S\|_{op} A)
$$

and thus, by combining the two cases, we find

$$
|d - d_n| \leq \log\det(\mathbb{I} + \|S_n^* S_n - S^* S\|_{op} A).
$$

Now, in order to upper bound $\|S^* S - S_n^* S_n\|_{op}$ with high probability, we use the following Bernstein inequality for a sum of random operators; see Rudi et al. [2015, Proposition 12] and Minsker [2017].

**Proposition S9** (Bernstein's inequality for a sum of i.i.d. random operators)**.** *Let $\mathcal{H}$ be a separable Hilbert space and let $X_1, \ldots, X_n$ be a sequence of independent and identically distributed self-adjoint positive random operators on $\mathcal{H}$. Assume that $\mathbb{E}X = 0$ and that there exists a real number $\ell > 0$ such that $\lambda_{\max}(X) \leq \ell$ almost surely. Let $\Sigma$ be a trace class positive operator such that $\mathbb{E}(X^2) \preceq \Sigma$. Then, for any $\delta \in (0, 1)$,*

$$
\lambda_{\max}\left(\frac{1}{n}\sum_{i=1}^n X_i\right) \leq \frac{2\ell\beta}{3n} + \sqrt{\frac{2\|\Sigma\|_{op}\beta}{n}}, \quad \text{where } \beta = \log\left(\frac{2\operatorname{Tr}(\Sigma)}{\|\Sigma\|_{op}\delta}\right),
$$

*with probability $1 - \delta$. If there further exists an $\ell'$ such that $\|X\|_{op} \leq \ell'$ almost surely, then, for any $\delta \in (0, 1/2)$,*

$$
\left\|\frac{1}{n}\sum_{i=1}^n X_i\right\|_{op} \leq \frac{2\ell'\beta}{3n} + \sqrt{\frac{2\|\Sigma\|_{op}\beta}{n}}, \quad \text{where } \beta = \log\left(\frac{2\operatorname{Tr}(\Sigma)}{\|\Sigma\|_{op}\delta}\right),
$$

*holds with probability $1 - 2\delta$.*

To apply Proposition S9 and conclude the proof, first recall the expression of the covariance operator

$$
C = S^* S = \int_{\mathcal{X}} \phi(x) \otimes \overline{\phi(x)}\mathrm{d}\mu(x),
$$

and of its sample version

$$
S_n^* S_n = \frac{1}{n}\sum_{i=1}^n \phi(x_i) \otimes \overline{\phi(x_i)}.
$$

Define $X_i = \phi(x_i) \otimes \overline{\phi(x_i)} - \int_{\mathcal{X}} \phi(x) \otimes \overline{\phi(x)} \mathrm{d}\mu(x)$. It is easy to check that $\mathbb{E} X_i = 0$ since $\mathbb{E}[\phi(x_i) \otimes \overline{\phi(x_i)}] = \int_{\mathcal{X}} \phi(x) \otimes \overline{\phi(x)} \mathrm{d}\mu(x)$. Then, using the triangle inequality and that $\sup_{x \in \mathcal{X}} k_{\mathcal{H}}(x, x) \leq \kappa^2$, we find the bound

$$
\begin{aligned}
\|X_i\|_{op} &= \left\| \int_{\mathcal{X}} \left( \phi(x_i) \otimes \overline{\phi(x_i)} - \phi(x) \otimes \overline{\phi(x)} \right) \mathrm{d}\mu(x) \right\|_{op} \\
&\leq \int_{\mathcal{X}} \left( \left\| \phi(x_i) \otimes \overline{\phi(x_i)} \right\|_{op} + \left\| \phi(x) \otimes \overline{\phi(x)} \right\|_{op} \right) \mathrm{d}\mu(x) \\
&\leq 2\kappa^2 \triangleq \ell'.
\end{aligned}
$$

Next, we compute a bound on the variance by bounding the second moment, namely

$$
\begin{aligned}
\mathbb{E}(X_i^2) &= \mathbb{E}\left[ \left( \phi(x_i) \otimes \overline{\phi(x_i)} \right)^2 \right] - \left( \mathbb{E}\left[ \phi(x_i) \otimes \overline{\phi(x_i)} \right] \right)^2 \\
&\preceq \mathbb{E}\left[ \left( \phi(x_i) \otimes \overline{\phi(x_i)} \right)^2 \right] \\
&\preceq \kappa^2 \mathbb{E}\left[ \phi(x_i) \otimes \overline{\phi(x_i)} \right] \triangleq \Sigma,
\end{aligned}
$$

where we used $\left( \phi(x_i) \otimes \overline{\phi(x_i)} \right)^2 = k_{\mathcal{H}}(x_i, x_i) \left( \phi(x_i) \otimes \overline{\phi(x_i)} \right)$. Also, we have

$$
\mathrm{Tr}(\Sigma) = \mathrm{Tr}\left( \kappa^2 \mathbb{E}\left[ \phi(x) \otimes \overline{\phi(x)} \right] \right) = \kappa^2 \int_{\mathcal{X}} k_{\mathcal{H}}(x, x) \mathrm{d}\mu(x) \leq \kappa^4.
$$

Moreover,

$$
\|\Sigma\|_{op} = \kappa^2 \left\| \mathbb{E}\left[ \phi(x) \otimes \overline{\phi(x)} \right] \right\|_{op} = \kappa^2 \lambda_{\max}(S^* S) = \kappa^2 \lambda_{\max}(\mathsf{T}_{k_{\mathcal{H}}}),
$$

where we used that $S^* S = \mathsf{T}_{k_{\mathcal{H}}}$ is the integral operator on $L^2(\mathcal{X})$ with integral kernel $k$. We are finally ready to apply Proposition S9. Since

$$
\beta \leq \log\left( \frac{2\kappa^2}{\lambda_{\max}(\mathsf{T}_{k_{\mathcal{H}}})\delta} \right),
$$

it holds, with probability at least $1 - 2\delta$,

$$
\|S^* S - S_n^* S_n\|_{op} \leq \frac{4\kappa^2 \log\left( \frac{2\kappa^2}{\lambda_{\max}(\mathsf{T}_{k_{\mathcal{H}}})\delta} \right)}{3n} + \sqrt{\frac{2\kappa^2 \lambda_{\max}(\mathsf{T}_{k_{\mathcal{H}}}) \log\left( \frac{2\kappa^2}{\lambda_{\max}(\mathsf{T}_{k_{\mathcal{H}}})\delta} \right)}{n}}.
$$

This concludes the proof of Theorem 1. $\qquad \square$

### S3.3 Statistical guarantee for the approximation of the log-likelihood by its sampled version

*Proof of Theorem 4.* The proof follows similar lines as in Rudi et al. [2020, Proof of Thm 5], with several adaptations. Let $\mathbf{B}_\star \in \mathcal{S}_+(\mathbb{R}^m)$ be the solution of (9). Notice that $A_\star$ and $V^* \mathbf{B}_\star V$ are distinct operators. Let $\bar{\mathbf{B}} = V A_\star V^* \in \mathcal{S}_+(\mathbb{R}^m)$ as in Lemma 2. Since $\mathbf{B}_\star$ has an optimal objective value, we have

$$
f_n(V^* \mathbf{B}_\star V) + \mathrm{Tr}(\lambda \mathbf{B}_\star) \leq f_n(V^* \bar{\mathbf{B}} V) + \mathrm{Tr}(\lambda \bar{\mathbf{B}}).
$$

Now we use the properties of $\bar{\mathbf{B}}$ given in Lemma 2, namely that $f_n(A_\star) = f_n(V^* \bar{\mathbf{B}} V)$ and $\mathrm{Tr}(\lambda \bar{\mathbf{B}}) \leq \mathrm{Tr}(\lambda A_\star)$. Then it holds

$$
f_n(V^* \bar{\mathbf{B}} V) + \mathrm{Tr}(\lambda \bar{\mathbf{B}}) \leq f_n(A_\star) + \mathrm{Tr}(\lambda A_\star).
$$

By combining the last two inequalities, we have $f_n(V^* \mathbf{B}_\star V) + \mathrm{Tr}(\lambda \mathbf{B}_\star) \leq f_n(A_\star) + \mathrm{Tr}(\lambda A_\star)$ and therefore

$$
f_n(V^* \mathbf{B}_\star V) - f_n(A_\star) \leq \mathrm{Tr}(\lambda A_\star) - \mathrm{Tr}(\lambda \mathbf{B}_\star). \tag{S5}
$$

We will use (S5) to derive upper and lower bounds on the gap $f(A_\star) - f_n(V^* \mathbf{B}_\star V)$.

**Lower bound.** Using Theorem 1, we have with high probability

$$|f(A_\star) - f_n(A_\star)| \leq \log \det(\mathbb{I} + c_n A),$$

and, in particular, this gives

$$f(A_\star) - f_n(A_\star) \leq \mathrm{Tr}(c_n A).$$

By combining this last inequality with (S5) and by using $\mathrm{Tr}(\lambda \mathbf{B}_\star) \geq 0$, we have the lower bound

$$\begin{aligned}
\Delta = f(A_\star) - f_n(V^* \mathbf{B}_\star V) = f(A_\star) - f_n(A_\star) + f_n(A_\star) - f_n(V^* \mathbf{B}_\star V) \\
\geq -\mathrm{Tr}(c_n A_\star) + \mathrm{Tr}(\lambda \mathbf{B}_\star) - \mathrm{Tr}(\lambda A_\star) \qquad \text{(S6)} \\
\geq -(c_n + \lambda) \mathrm{Tr}(A_\star).
\end{aligned}$$

**Upper bound.** We have the bound with high probability (for the same event as above)

$$\Delta = f(A_\star) - f_n(\mathbf{B}_\star) = \underbrace{f(A_\star) - f(V^* \mathbf{B}_\star V)}_{\leq 0} + f(V^* \mathbf{B}_\star V) - f_n(V^* \mathbf{B}_\star V)$$

$$\begin{aligned}
\leq \log \det(\mathbb{I} + c_n V^* \mathbf{B}_\star V) \quad \text{(Theorem 1)} \\
\leq \mathrm{Tr}(c_n V V^* \mathbf{B}_\star) \quad \text{(cyclicity)} \\
\leq \mathrm{Tr}(c_n \mathbf{B}_\star). \quad \text{(since } V V^* \text{ is a projector)} \qquad \text{(S7)}
\end{aligned}$$

By combining this with (S6), we find

$$-\mathrm{Tr}(c_n A_\star) + \mathrm{Tr}(\lambda \mathbf{B}_\star) - \mathrm{Tr}(\lambda A_\star) \leq \mathrm{Tr}(c_n \mathbf{B}_\star).$$

Since, by assumption, we have $c_n \leq \lambda - c_n$, the latter inequality becomes

$$c_n \mathrm{Tr}(\mathbf{B}_\star) \leq (c_n + \lambda) \mathrm{Tr}(A_\star). \qquad \text{(S8)}$$

By using (S8) in the bound in (S7), we obtain

$$f(A_\star) - f_n(V^* \mathbf{B}_\star V) \leq (c_n + \lambda) \mathrm{Tr}(A_\star),$$

Thus, the upper and lower bound yield together

$$|f(A_\star) - f_n(V^* \mathbf{B}_\star V)| \leq (c_n + \lambda) \mathrm{Tr}(A_\star) \leq \left(\frac{\lambda}{2} + \lambda\right) \mathrm{Tr}(A_\star),$$

where we used once more $2c_n \leq \lambda$. This is the desired result. □

*Proof of Corollary 5.* We use the triangle inequality

$$|f(A_\star) - f(V \mathbf{B}_\star V^*)| \leq |f(A_\star) - f_n(V \mathbf{B}_\star V^*)| + |f_n(V \mathbf{B}_\star V^*) - f(V \mathbf{B}_\star V^*)|.$$

The first term is upper bounded whp by Theorem 4. The second term is bounded by Theorem 1 as follows

$$|f_n(V \mathbf{B}_\star V^*) - f(V \mathbf{B}_\star V^*)| \leq \mathrm{Tr}(c_n \mathbf{B}_\star)$$

with $\mathrm{Tr}(c_n \mathbf{B}_\star) \leq \frac{3}{2} \lambda \mathrm{Tr}(A_\star)$ as in (S8) in the proof of Theorem 4. □

### S3.4 Numerical approach: convergence study

*Proof of Theorem 3.* This proof follows the same technique as in Mariet and Sra [2015], with some extensions. Let $\Sigma = \mathbf{X}^{-1}$. We decompose the objective

$$g(\Sigma^{-1}) = \log \det \left(\mathbf{I}_m + \frac{1}{|\mathcal{I}|} \Sigma^{-1}\right)_{\mathcal{II}} - \frac{1}{s} \sum_{\ell=1}^{s} \log \det(\mathbf{U}_\ell^\top \Sigma^{-1} \mathbf{U}_\ell) + \lambda \mathrm{Tr}(\Sigma^{-1} \mathbf{K}^{-1})$$

as the following sum: $g(\Sigma^{-1}) = h_1(\Sigma) + h_2(\Sigma)$, where $h_1(\Sigma) = -\log \det(\Sigma) + \lambda \mathrm{Tr}(\Sigma^{-1} \mathbf{K}^{-1})$ is a strictly convex function on $\Sigma \succ 0$ and

$$h_2(\Sigma) = \log \det(\Sigma) + \log \det \left(\mathbf{I}_m + \frac{1}{|\mathcal{I}|} \Sigma^{-1}\right)_{\mathcal{II}} - \frac{1}{s} \sum_{\ell=1}^{s} \log \det(\mathbf{U}_\ell^\top \Sigma^{-1} \mathbf{U}_\ell)$$

is concave on $\mathbf{\Sigma} \succ 0$. We refer to Lemma S2 and S3 for the former and latter statements, respectively. Then, we use the concavity of $h_2$ to write the following upper bound

$$h_2(\mathbf{\Sigma}) \leq h_2(\mathbf{Y}) + \mathrm{Tr}\left(\nabla h_2(\mathbf{Y})(\mathbf{\Sigma} - \mathbf{Y})\right),$$

where the matrix-valued gradient is

$$\nabla h_2(\mathbf{Y}) = \mathbf{Y}^{-1} - \mathbf{Y}^{-1}\mathbf{U}_{\mathcal{I}}\left(|\mathcal{I}|\mathbf{I}_{|\mathcal{I}|} + \mathbf{U}_{\mathcal{I}}^{\top}\mathbf{Y}^{-1}\mathbf{U}_{\mathcal{I}}\right)^{-1}\mathbf{U}_{\mathcal{I}}^{\top}\mathbf{Y}^{-1}$$
$$+ \frac{1}{s}\sum_{\ell=1}^{s}\mathbf{Y}^{-1}\mathbf{U}_{\ell}\left(\mathbf{U}_{\ell}^{\top}\mathbf{Y}^{-1}\mathbf{U}_{\ell}\right)^{-1}\mathbf{U}_{\ell}^{\top}\mathbf{Y}^{-1}.$$

Define the auxillary function $\xi(\mathbf{\Sigma}, \mathbf{Y}) \triangleq h_1(\mathbf{\Sigma}) + h_2(\mathbf{Y}) + \mathrm{Tr}\left(\nabla h_2(\mathbf{Y})(\mathbf{\Sigma} - \mathbf{Y})\right)$ which satisfies $g(\mathbf{\Sigma}^{-1}) \leq \xi(\mathbf{\Sigma}, \mathbf{Y})$ and $g(\mathbf{\Sigma}^{-1}) = \xi(\mathbf{\Sigma}, \mathbf{\Sigma})$. We define the iteration $\mathbf{X}_k = \mathbf{\Sigma}_k^{-1}$ where

$$\mathbf{\Sigma}_{k+1} = \arg\min_{\mathbf{\Sigma}\succ 0}\xi(\mathbf{\Sigma}, \mathbf{\Sigma}_k),$$

so that it holds $g(\mathbf{\Sigma}_{k+1}^{-1}) \leq \xi(\mathbf{\Sigma}_{k+1}, \mathbf{\Sigma}_k) \leq \xi(\mathbf{\Sigma}_k, \mathbf{\Sigma}_k) = g(\mathbf{\Sigma}_k^{-1})$. Thus, this iteration has monotone decreasing objectives. It remains to show that this iteration corresponds to (13). The solution of the above minimization problem can be obtained by solving the first order optimality condition since $\xi(\cdot, \mathbf{Y})$ is strictly convex. This gives

$$-\mathbf{\Sigma}^{-1} - \lambda\mathbf{\Sigma}^{-1}\mathbf{K}^{-1}\mathbf{\Sigma}^{-1} + \mathbf{\Sigma}_k^{-1} - \mathbf{\Sigma}_k^{-1}\mathbf{U}_{\mathcal{I}}\left(|\mathcal{I}|\mathbf{I}_{|\mathcal{I}|} + \mathbf{U}_{\mathcal{I}}^{\top}\mathbf{\Sigma}_k^{-1}\mathbf{U}_{\mathcal{I}}\right)^{-1}\mathbf{U}_{\mathcal{I}}^{\top}\mathbf{\Sigma}_k^{-1}$$
$$+ \frac{1}{s}\sum_{\ell=1}^{s}\mathbf{\Sigma}_k^{-1}\mathbf{U}_{\ell}\left(\mathbf{U}_{\ell}^{\top}\mathbf{\Sigma}_k^{-1}\mathbf{U}_{\ell}\right)^{-1}\mathbf{U}_{\ell}^{\top}\mathbf{\Sigma}_k^{-1} = 0.$$

Now, we replace $\mathbf{X} = \mathbf{\Sigma}^{-1}$ in the above condition. After a simple algebraic manipulation, we obtain the following condition

$$\mathbf{X} + \lambda\mathbf{X}\mathbf{K}^{-1}\mathbf{X} = p(\mathbf{X}_k),$$

where, as defined in (13), $p(\mathbf{X}) = \mathbf{X} + \mathbf{X}\mathbf{\Delta}\mathbf{X}$ and

$$\mathbf{\Delta} = \frac{1}{s}\sum_{\ell=1}^{s}\mathbf{U}_{\ell}\mathbf{X}_{\mathcal{C}_{\ell}\mathcal{C}_{\ell}}^{-1}\mathbf{U}_{\ell}^{\top} - \mathbf{U}_{\mathcal{I}}\left(|\mathcal{I}|\mathbf{I}_{|\mathcal{I}|} + \mathbf{U}_{\mathcal{I}}^{\top}\mathbf{X}\mathbf{U}_{\mathcal{I}}\right)^{-1}\mathbf{U}_{\mathcal{I}}^{\top}.$$

Finally, we introduce the Cholesky decomposition $\mathbf{K} = \mathbf{R}^{\top}\mathbf{R}$, so that we have an equivalent identity

$$\mathbf{R}^{-1\top}\mathbf{X}\mathbf{R}^{-1} + \lambda\left(\mathbf{R}^{-1\top}\mathbf{X}\mathbf{R}^{-1}\right)^2 - \mathbf{R}^{-1\top}p(\mathbf{X}_k)\mathbf{R}^{-1} = 0.$$

Let $\mathbf{X}' = \mathbf{R}^{-1\top}\mathbf{X}\mathbf{R}^{-1}$ and $p'(\mathbf{X}_k) = \mathbf{R}^{-1\top}p(\mathbf{X}_k)\mathbf{R}^{-1}$. The positive definite solution of this second order matrix equation write

$$\mathbf{X}' = \frac{-\mathbf{I}_m + (\mathbf{I}_m + 4\lambda p'(\mathbf{X}_k))^{1/2}}{2\lambda},$$

which directly yields (13). $\qquad\square$

### S3.5 Approximation of the correlation kernel

We start by proving the following useful result.

**Theorem S10** (Correlation kernel approximation, formal version). *Let $\delta \in (0,1]$ be the failure probability and let $\gamma > 0$ be a scale factor. Let $\hat{\mathsf{K}}(\gamma)$ be the approximation defined in (11) with i.i.d. sampling of $p$ points. Let $p$ be large enough so that $t(p) > 1$ with $t(p) = \frac{4c^2\beta}{3\gamma p} + \sqrt{\frac{2c^2\beta}{\gamma p}}$ where $c^2 = \kappa^2\|A\|_{op}$ and $\beta = \log\left(\frac{4d_{\mathrm{eff}}(\gamma)}{\delta\|\mathsf{K}(\gamma)\|_{op}}\right)$ Then, with probability $1 - \delta$ it holds that*

$$\frac{1}{1 + t(p)}\mathsf{K}(\gamma) \preceq \hat{\mathsf{K}}(\gamma) \preceq \frac{1}{1 - t(p)}\mathsf{K}(\gamma).$$

*Furthermore, if we assume $\gamma \leq \lambda_{\max}(\mathsf{A})$, we can take $\beta = \log\left(\frac{8d_{\mathrm{eff}}(\gamma)}{\delta}\right) \leq \frac{8d_{\mathrm{eff}}(\gamma)}{\delta}$.*

*Proof.* For simplicity, define $\Psi : \mathcal{H} \to \mathbb{R}^m$ as $\Psi = \sqrt{m}\mathbf{\Lambda}S_m$, which is such that $A_\star = \Psi^*\Psi$. Then, we write

$$\hat{\mathsf{K}} = S\Psi^*(\Psi S_p^* S_p \Psi^* + \gamma\mathbf{I}_m)^{-1}\Psi S^*,$$

where we recall that $S_p^* S_p = \frac{1}{p}\sum_{i=1}^p \phi(x_i'') \otimes \overline{\phi(x_i'')}$. Next, we used the following result.

**Proposition S11** (Proposition 5 in Rudi et al. [2018] with minor adaptations)**.** *Let $\gamma > 0$ and $v_1, \ldots, v_p$ with $p \geq 1$ be identically distributed random vectors on a separable Hilbert space $H$, such that there exists $c^2 > 0$ for which $\|v\|_H \leq c^2$ almost surely. Denote by $Q$ the Hermitian operator $Q = \frac{1}{p}\sum_{i=1}^p \mathbb{E}[v_i \otimes \overline{v_i}]$. Let $Q_p = \frac{1}{p}\sum_{i=1}^p v_i \otimes \overline{v_i}$. Then for any $\delta \in (0, 1]$, the following holds*

$$\|(Q + \gamma\mathbb{I})^{-1/2}(Q - Q_p)(Q + \gamma\mathbb{I})^{-1/2}\|_{op} \leq \frac{4c^2\beta}{3\gamma p} + \sqrt{\frac{2c^2\beta}{\gamma p}},$$

*with probability $1 - \delta$ and $\beta = \log\frac{4\,\mathrm{Tr}\left(Q(Q+\gamma\mathbb{I})^{-1}\right)}{\delta\|Q(Q+\gamma\mathbb{I})^{-1}\|_{op}} \leq 8\frac{c^2/\|Q\|_{op} + \mathrm{Tr}\left(Q(Q+\gamma\mathbb{I})^{-1}\right)}{\delta}$.*

Then, we merely define the following vector $\mathbf{v}_i = \Psi\phi(x_i'')$ for $1 \leq i \leq p$ so that

$$\mathbf{Q}_p = \Psi S_p^* S_p \Psi^* = \frac{1}{p}\sum_{i=1}^p \Psi\left(\phi(x_i'') \otimes \overline{\phi(x_i'')}\right)\Psi^*.$$

Furthermore, we define $\mathbf{Q} = \Psi S^* S\Psi^*$. Also, we have $S^*S = \int_\mathcal{X} \phi(x) \otimes \overline{\phi(x)}\mathrm{d}\mu(x)$, so that $\mathbb{E}[S_p^* S_p] = S^*S$. Hence, it holds $\mathbb{E}[\mathbf{Q}_p] = \mathbf{Q}$. First, by using $\Psi^*\Psi \preceq \|\Psi^*\Psi\|_{op}\mathbb{I}$, we have

$$\|\mathbf{v}\|_2^2 = \langle\phi(x), \Psi^*\Psi\phi(x)\rangle \leq k_\mathcal{H}(x, x)\|\Psi^*\Psi\|_{op} \leq \kappa^2\|\Psi^*\Psi\|_{op} = \kappa^2\|A_\star\|_{op},$$

almost surely. Next, we calculate the following quantity

$$\begin{aligned}
\mathrm{Tr}\left[\mathbf{Q}(\mathbf{Q} + \gamma\mathbf{I}_m)^{-1}\right] &= \mathrm{Tr}\left[\Psi S^* S\Psi^*(\Psi S^* S\Psi^* + \gamma\mathbf{I}_m)^{-1}\right] \\
&= \mathrm{Tr}\left[S\Psi^*(\Psi S^* S\Psi^* + \gamma\mathbf{I}_m)^{-1}\Psi S^*\right] \\
&= \mathrm{Tr}\left[S\Psi^*\Psi S^*(S\Psi^*\Psi S^* + \gamma\mathbb{I})^{-1}\right] \\
&= \mathrm{Tr}\left[A(A + \gamma\mathbb{I})^{-1}\right],
\end{aligned}$$

where we used the push-through identity at the next to last equality.

For obtaining the bound on $\beta$, we first write

$$\|\mathbf{Q}(\mathbf{Q} + \gamma\mathbf{I}_m)^{-1}\|_{op} = \|A(A + \gamma\mathbb{I})^{-1}\|_{op} = (1 + \gamma/\lambda_{\max}(A))^{-1}.$$

To lower bound the latter quantity we require $\gamma \leq \lambda_{\max}(A)$ and hence $\|\mathbf{Q}(\mathbf{Q} + \gamma\mathbf{I}_m)^{-1}\|_{op} \geq 1/2$. For the remainder of the proof, we show the main matrix inequality. For convenience, define the upperbound in Proposition S11 as

$$t(p) = \frac{4c^2\beta}{3\gamma p} + \sqrt{\frac{2c^2\beta}{\gamma p}}. \tag{S9}$$

Thanks to Proposition S11, we know that with probability $1 - \delta$, we have

$$-t\left(\Psi SS^*\Psi^* + \gamma\mathbf{I}_m\right) \preceq \Psi SS^*\Psi^* - \Psi S_p^* S_p\Psi^* \preceq t\left(\Psi SS^*\Psi^* + \gamma\mathbf{I}_m\right),$$

or equivalently

$$\Psi SS^*\Psi^* - t(\Psi SS^*\Psi^* + \gamma\mathbf{I}_m) \preceq \Psi S_p^* S_p\Psi^* \preceq \Psi SS^*\Psi^* + t(\Psi SS^*\Psi^* + \gamma\mathbf{I}_m).$$

By simply adding $\gamma\mathbf{I}_m$ to these inequalities, we obtain

$$(1 - t)(\Psi SS^*\Psi^* + \gamma\mathbf{I}_m) \preceq \Psi S_p^* S_p\Psi^* + \gamma\mathbf{I}_m \preceq (1 + t)(\Psi SS^*\Psi^* + \gamma\mathbf{I}_m).$$

Hence, if $t < 1$, by a simple manipulation, we find

$$(1 + t)^{-1}(\Psi SS^*\Psi^* + \gamma\mathbf{I}_m)^{-1} \preceq (\Psi S_p^* S_p\Psi^* + \gamma\mathbf{I}_m)^{-1} \preceq (1 - t)^{-1}(\Psi SS^*\Psi^* + \gamma\mathbf{I}_m)^{-1}.$$

By acting with $S\Psi^*$ on the left and $\Psi S^*$ on the right, and then, using the push-through identity

$$S\Psi^*(\Psi SS^*\Psi^* + \gamma\mathbf{I}_m)^{-1}\Psi S^* = (S\Psi^*\Psi S + \gamma\mathbb{I})^{-1}S^*\Psi^*\Psi S^*,$$

the desired result follows. $\qquad\square$

We can now prove Theorem 6 by simplifying some of the bounds given in Theorem S10.

*Proof of Theorem 6.* Consider the upper bound given in (S9). We will simplify it to capture the dominant behavior as $p \to +\infty$. Assume $\sqrt{\frac{2c^2\beta}{\gamma p}} < 1$, or equivalently $p > \frac{2c^2\beta}{\gamma}$. In this case, we give a simple upper bound on $t(p)$ as follows

$$t(p) < \left(\frac{2}{3} + 1\right)\sqrt{\frac{2c^2\beta}{\gamma p}} < \sqrt{\frac{8c^2\beta}{\gamma p}},$$

so that we avoid manipulating cumbersome expressions. Thus, if we want the latter bound to be smaller than $\epsilon \in (0, 1)$, we require

$$p \geq \frac{8c^2\beta}{\gamma\epsilon^2},$$

which is indeed larger than $\frac{2c^2\beta}{\gamma}$ since $1/\epsilon > 1$. Thus, by using the same arguments as in the proof of Theorem S10, we have the multiplicative error bound

$$\frac{1}{1+\epsilon}\mathsf{K}(\gamma) \preceq \hat{\mathsf{K}}(\gamma) \preceq \frac{1}{1-\epsilon}\mathsf{K}(\gamma),$$

with probability at least $1 - \delta$ if

$$p \geq \frac{8c^2\beta}{\gamma\epsilon^2} = \frac{8\kappa^2\|A\|_{op}}{\gamma\epsilon^2}\log\left(\frac{4d_{\text{eff}}(\gamma)}{\delta\|\mathsf{K}\|_{op}}\right)$$

where the last equality is obtained by substituting $c^2 = \kappa^2\|A\|_{op}$ and $\beta = \log\left(\frac{4d_{\text{eff}}(\gamma)}{\delta\|\mathsf{K}(\gamma)\|_{op}}\right)$ given in Theorem S10. □

## S4 Supplementary empirical results

### S4.1 Finer analysis of the Gaussian L-ensemble estimation problem of Section 6

In this section, we report results corresponding to the simulation setting of Section 6 with $\rho = 100$.

**Intensity estimation from several DPP samples.** In Figure S1, we replicate the setting of Figure 1 with $s = 3$ and $s = 10$ DPP samples and a smaller regularization parameter. The estimated intensity is then closer to the ground truth ($\rho = 100$) for a large value of $s$, although there are small areas of high intensity at the boundary of the domain $[0, 1]^2$. A small improvement is also observed by increasing $s$ from 3 (left) to 10 (right), namely the variance of the estimated intensity tends to decrease when $s$ increases. In Figure S2, we illustrate the intensity estimation in the case of a large and small $\sigma$, respectively on the left and right columns. As expected, a large value of $\sigma$ has a regularization effect but also leads to an underestimation of the intensity. On the contrary, a small value of $\sigma$ seems to cause inhomogeneities in the estimated intensity.

**Correlation structure estimation.** The general form of the correlation kernel is also important. In order to visualize the shape of the correlation kernel $\hat{\mathsf{k}}(x, y)$, we display in Figure S3 the Gram matrices of the estimated $[\hat{\mathsf{k}}(x, x')]_{x,x'\in\text{grid}}$ and ground truth correlation kernels $[\mathsf{k}(x, x')]_{x,x'\in\text{grid}}$ on a square grid, for the same parameters as in Figure S1 (RHS). After removing the boundary effects, we observe that the estimated correlation kernel shape closely resembles the ground truth although the decay of the estimated kernel seems to be a bit slower. Moreover, we observe some 'noise' in the tail of the estimated kernel. Again, the intensity of the estimated process is also a bit underestimated.

In the context of point processes, it is common to compute summary statistics from samples to 'understand' the correlation structure of a stationary process. It is strictly speaking not possible to calculate e.g. Ripley's K function (see Baddeley et al. [2015]) since our estimated correlation kernel is not stationary, that is, there exits no function $t(\cdot)$ such that $\hat{\mathsf{k}}(x, y) = t(x - y)$.

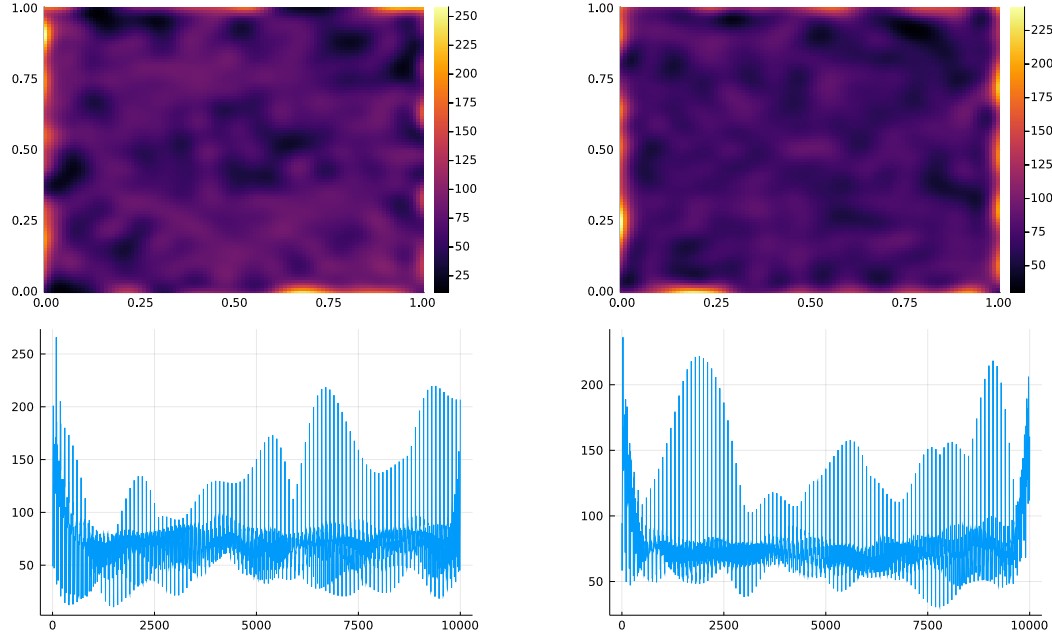

Figure S1: Effet of the number of samples on intensity estimation (with $\sigma = 0.1$ and $\lambda = 10^{-4}$, as in Figure 1). Left column: estimation from $s = 3$ DPP samples. Right column: estimation from $s = 10$ DPP samples. The first row is a heatmap of the intensity $\hat{\mathsf{k}}(x, x)$ on a $100 \times 100$ grid within $[0, 1]^2$. The second row is the same data matrix in a flattened format, that is, each column of the $100 \times 100$ data matrix is concatenated to form a $10000 \times 1$ matrix whose entries are plotted. Notice that the sharp peaks are due to boundary effects. These peaks are regularly spaced due to the column-wise unfolding.

## S4.2 Convergence of the regularized Picard iteration

In particular, we illustrate the convergence of the regularized Picard iteration to the exact solution given in Proposition S6. In Figure S4, we solve the problem (6) in the case $\mathcal{I} = \mathcal{C}$ with $s = 1$ where $\mathcal{C}$ is the unique DPP sample. For simplicity, we select the DPP sample used in Figure 1 ($\rho = 100$, bottom row). This illustrates that the regularized Picard iteration indeed converges to the unique solution in this special case.

## S4.3 Complexity and computing ressources

**Complexity.** The space complexity of our method is dominated by the space complexity of storing the kernel matrix $\mathbf{K}$ in Algorithm 1, namely $O(m^2)$ where we recall that $m = |\mathcal{Z}|$ with $\mathcal{Z} \triangleq \cup_{\ell=1}^{s} \mathcal{C}_\ell \cup \mathcal{I}$. The time complexity of one iteration of (14) is dominated by the matrix square root, which is similar to the eigendecomposition, i.e., $O(m^3)$. The time complexity of Algorithm 2 is dominated by the Cholesky decomposition and the linear system solution, i.e., $O(m^3)$.

**Computing ressources.** A typical computation time is 65 minutes to solve the example of Figure 1 (bottom row) on 8 virtual cores of a server with two 18 core Intel Xeon E5-2695 v4s (2.1 Ghz). The computational bottleneck is the regularized Picard iteration.

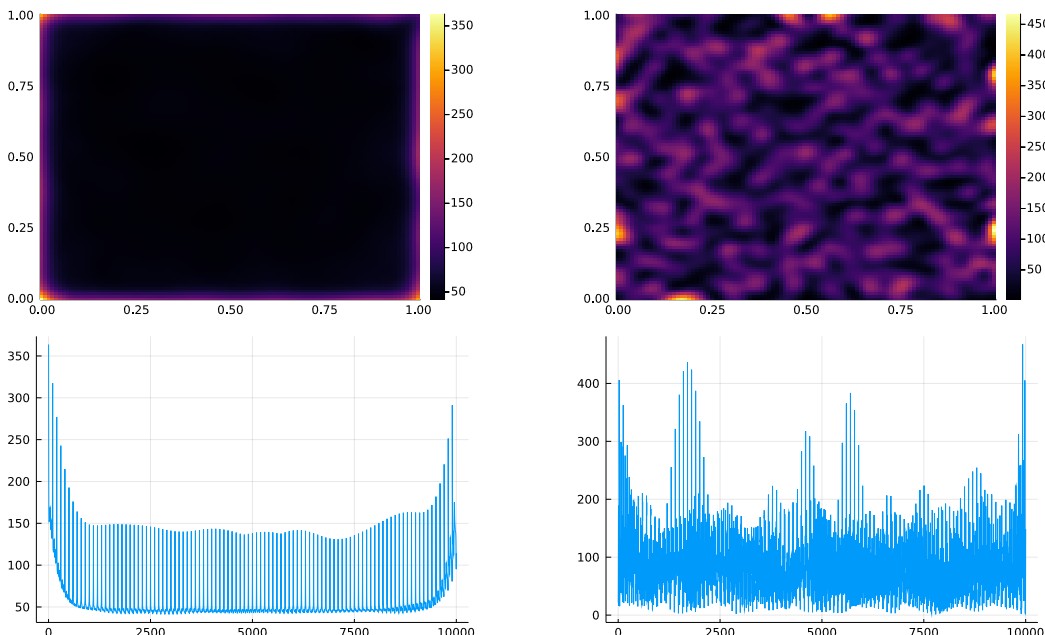

Figure S2: Effet of the bandwidth $\sigma$ on intensity estimation. Left column: large value ($\sigma = 0.15$, $\lambda = 10^{-4}$) with $s = 10$ DPP samples. Right column: small value ($\sigma = 0.05$, $\lambda = 10^{-4}$) with $s = 3$ DPP samples. The first row is a heatmap of the intensity on $[0, 1]^2$. The second row is the same data matrix in a flattened format, that is, each column of the $100 \times 100$ data matrix is concatenated to form a $10000 \times 1$ matrix whose entries are plotted. Notice that the sharp peaks at the bottom row are due to boundary effects. These peaks are regularly spaced due to the column-wise unfolding.

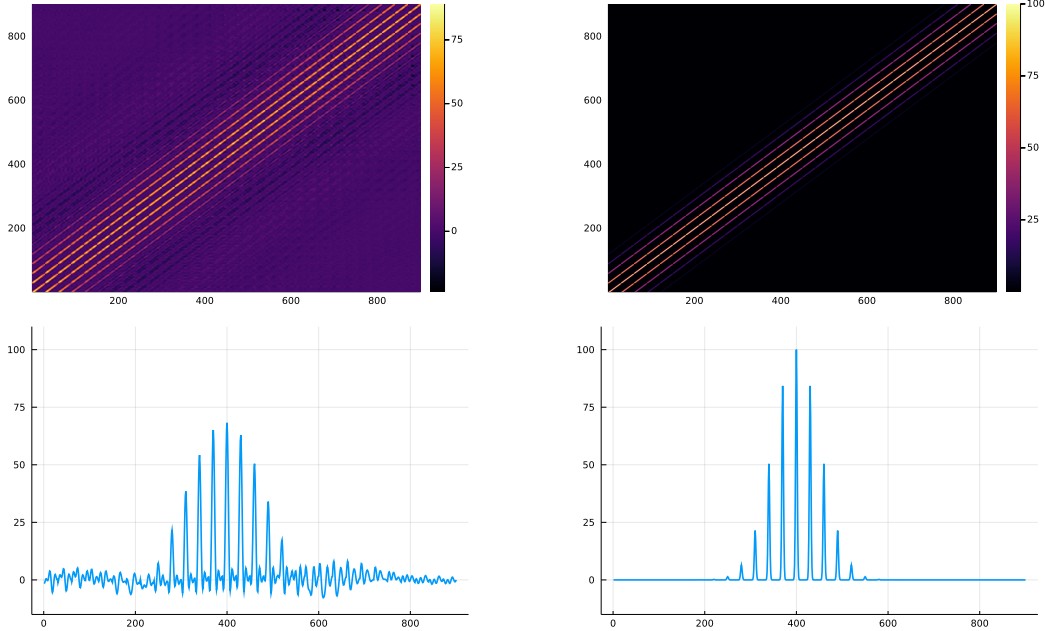

Figure S3: Correlation kernel estimation without boundary effect. We display a Gram matrix of the estimated correlation kernel $[\hat{\mathsf{k}}(x,x')]_{x,x'\in\text{grid}}$ (left column) and ground truth correlation kernel $[\mathsf{k}(x,x')]_{x,x'\in\text{grid}}$ (right column) on a $30\times 30$ grid within $[0.2,0.8]^2$ for the example of Figure S1 with $s=10$, $\sigma=0.1$ and $\lambda=10^{-4}$. The first row is a heatmap of the Gram matrices, while the second row is a one-dimensional slice of the above Gram matrices at index 400. This second row of plots allows to visually compare the bell shape of the approximate and exact correlation kernels. The apparent discontinuities in the Gaussian kernel shape are an artifact due to the manner the grid points are indexed. Notice that the correlation kernels are evaluated on a smaller domain within $[0,1]^2$ in order to remove boundary effects.

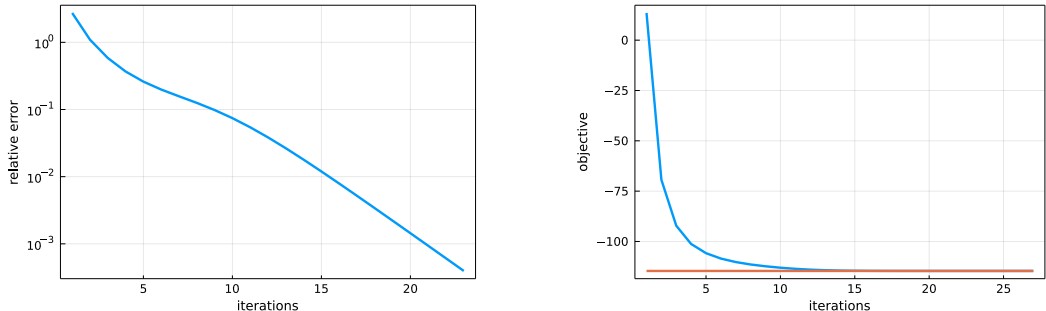

Figure S4: Convergence towards the exact solution for the example of Figure 1 (bottom row) with the parameters $\sigma=0.1$ and $\lambda=0.1$. Left: Relative error with the exact solution in Frobenius norm $\|\mathbf{B}-\mathbf{B}_{\text{exact}}\|_F/\|\mathbf{B}_{\text{exact}}\|_F$ w.r.t. the iteration number. Right: Objective value (blue line) and optimal objective (red line) vs iteration number. The stopping criterion is here tol $=10^{-7}$.