# OpenReview forum: "Nonparametric estimation of continuous DPPs with kernel methods"
_NeurIPS.cc/2021/Conference — NeurIPS 2021 Poster_

### Official Review · Reviewer_UmsC · 2021-07-09

**Rating:** 7
**Confidence:** 3

**Summary:**

This paper presents a new approach for maximum likelihood estimation (MLE) for continuous determinantal point processes (DPPs), which leverages recent work on kernel methods.  This approach involves nonparametric learning for a restricted class of the MLE problem, which is shown to fall within the scope of recent work on a representer theorem for nonnegative functions in an RKHS.  An optimization problem over matrices is characterized, and then a fixed point algorithm for solving the finite-dimensional optimization problem is proposed.  Statistical guarantees for the MLE approximation are presented.  Finally, the authors provide an empirical validation of their proposed method using experiments run on synthetic data sampled from a DPP.

**Limitations And Societal Impact:**

The authors have included a short, but adequate, discussion of the limitations of their work in Section 7 of the paper.  Since this work involves very general algorithmic and theoretical contributions, a discussion of potential negative societal impact does not appear to be applicable.

**Main Review:**

The MLE learning approach for continuous DPPs proposed in this paper is novel, and appears to be a significant contribution.  As noted by the authors, substantial prior work exists on discrete or finite DPPs, but much less attention has been paid to continuous DPPs within the machine learning community.  This paper is reasonably well written, and appears to be one of the first contributions in the area of nonparametric learning for continuous DPPs.  The main theoretical contributions in Sections 3 - 5 of the paper appear to be sound, although I have not carefully checked the proofs.  Furthermore, the statistical guarantees in Section 5 establish approximation bounds on the MLE solution and correlation kernel that appear to be reasonably tight under realistic assumptions, with the exception of the lower bound on $p$ in Theorem 6.

A few aspects of this paper could be improved.  First, the authors should discuss the computational complexity of the algorithms for estimating the DPP likelihood kernel (Alg. 1) and correlation kernel (Alg. 2).  What is the computational complexity of the fixed point algorithm in Sec. 4?  A very brief discussion of space and time complexity is given in Appendix S4.3, but this discussion should be significantly expanded upon.  Second, experiments that involve using the proposed approach to learn a continuous DPP from a real dataset should be performed.  Lastly, it would be helpful for the authors to provide a more detailed comparison of their nonparametric learning approach with prior work on parametric learning for continuous DPPs.

**Time Spent Reviewing:**

5 hours

---

> ### Author Response · Authors · 2021-08-10
> **Answer to Reviewer UmsC**
>
> Thank you for your review.
> > A few aspects of this paper could be improved. First, the authors should discuss the computational complexity of the algorithms for estimating the DPP likelihood kernel (Alg. 1) and correlation kernel (Alg. 2). What is the computational complexity of the fixed point algorithm in Sec. 4?
> A very brief discussion of space and time complexity is given in Appendix S4.3, but this discussion should be significantly expanded upon.
>
> We will add to S4.3 a discussion of the space and time complexity of the operations in Alg. 1 and Alg. 2. They are dominated by the Cholesky decomposition and linear system inversion.
> > Second, experiments that involve using the proposed approach to learn a continuous DPP from a real dataset should be performed. Lastly, it would be helpful for the authors to provide a more detailed comparison of their nonparametric learning approach with prior work on parametric learning for continuous DPPs.
>
> Thank you for this suggestion. So far, the main contribution of our paper is theoretical. We meant our simulations to both support the feasibility of our approach and investigate the effect of several hyperparameters. We are working on a comparison with parametric approaches on benchmark datasets [Lavancier et al., JRSSB'15 in our references], but setting up a fair and intelligible comparison requires more methodological work first. For instance, most work on parametric inference for continuous DPPs deals with stationary DPPs and rectangular windows. From a parametric point of view, it is a natural restriction, because one can then diagonalize the kernel in the Fourier basis, and approximate the likelihood. On the positive side, our nonparametric approach does not suffer from any such limitation: we can infer nonstationary kernels and use very irregular observation windows. On the negative side, we have no way yet to constrain our approach to stationary kernels. Using a standard benchmark for stationary kernels would thus see our method infer a nonstationary kernel (although, as seen in our experimental section, the output would likely be only mildly non-stationary). Common measures of divergence between stationary kernels would thus not apply, for instance.

---

> > ### Comment · Reviewer_UmsC · 2021-08-17
> > **Rebuttal response**
> >
> > Thank you for the detailed response to my questions/comments.  I’ve read through the other reviews and rebuttal comments, and am satisfied with the remarks.  I maintain my review score of 7 (Good paper, accept).

---

### Official Review · Reviewer_jwGL · 2021-07-16

**Rating:** 7
**Confidence:** 3

**Summary:**

This paper introduces an algorithm to learn non-parametric, continuous determinantal point processes (DPPs). This algorithm relies on recent representer results for nonnegative functions in a RKHS and on a previous, related algorithm to learn discrete DPP parameterizations.

The authors prove that the proposed algorithm is guaranteed to increase the MLE objective function at each iteration, as well as prove statistical guarantees for the resulting solution: distance to an optimal parameterization, sample complexity.

**Limitations And Societal Impact:**

The authors clearly describe the limitations of their work; I do not see a societal impact that requires discussing.

**Main Review:**

Originality: The proposed method to learn a non-parameteric, continuous DPP is new; it is, to the extent of my knowledge, the first result on learning non-parametric, continuous DPPs.

Quality: This paper is of high quality; the proofs are detailed, the theory meticulous (to the extent that I can judge it, being unfamiliar with results in Marteau-Ferey et al., 2020, and Rudi et al., 2020.). Furthermore, the theoretical analysis is regularly tied back to core concepts in DPPs (e.g., the effective sample size), and key arguments are regularly highlighted to support the provided analysis.

Clarity: With the caveat that I am not familiar with continuous DPPs (but familiar with their discrete counterparts), I found this paper very well written and accessible. The authors carefully frame their contributions in terms of previously known results within continuous and discrete DPP landscapes, draw the reader's attention to crucial arguments in their proofs, and describe in detail the assumptions and potential limitations of their work. Overall, this is a very well written paper.

Significance: The results are significant to the community, in particular given growing interest in the applications of continuous DPPs. This is the first result on learning non-parametric continuous DPPs, and also has interesting implications for potential extensions to learning algorithms for discrete DPPs. One potential limitation of this work is that the empirical verification focuses only on exponentiated quadratic kernels.


Questions & Comments:
- You allude in the final section that the regularized objective might transfer to the finite DPP setting. Do you mean that by introducing a trace-based regularizer to the discrete Picard iteration, you may be able to recover a guarantee of the style of Theorem 4?
- Have you tried learning L-ensembles with correlation kernels that are not of the same form as the RKHS kernel? Would you expect this to influence the learning process?

**Time Spent Reviewing:**

5 hours

---

> ### Author Response · Authors · 2021-08-10
> **Answer to Reviewer jwGL**
>
> Thank you for your review.
> > You allude in the final section that the regularized objective might transfer to the finite DPP setting. Do you mean that by introducing a trace-based regularizer to the discrete Picard iteration, you may be able to recover a guarantee of the style of Theorem 4?
>
> Indeed, this is one avenue. Similarly, a trace-like regularization could be a good "prior" (in the PAC-Bayes sense) to yield good frequentist guarantees for the corresponding Bayesian estimator. This could help when the dataset is small.
> > Have you tried learning L-ensembles with correlation kernels that are not of the same form as the RKHS kernel? Would you expect this to influence the learning process?
>
> This is a very good question. In particular, we believe that the choice of the RKHS should be governed both by prior knowledge on the smoothness of the underlying kernel, but also by the properties of the observation window. For observation windows with nonlinear boundaries, for instance, Sobolev kernels are promising, especially in the light of [Rudi et al., Regularized kernel algorithms for support estimation, Frontiers in Applied Mathematics and Statistics 3, 23, 2017].

---

> > ### Comment · Reviewer_jwGL · 2021-08-26
> > **Thank you for the rebuttal**
> >
> > I thank the authors for their rebuttal; my score remains unchanged (7) and I recommend accepting this paper.

---

### Official Review · Reviewer_dbvc · 2021-07-21

**Rating:** 7
**Confidence:** 3

**Summary:**

This paper studies nonparametric MLE inference of continuous determinantal point processes (DPPs). The authors analyze various representations between discrete DPPs and continuous DPPs. In particular, they observe that the difference of normalization of DPP is bounded by the trace of a matrix. Then, they consider a new optimization problem penalizing the trace and propose a method of estimating likelihood/correlation DPP kernels. Moreover, they study an iterative method (a.k.a Picard iteration) for solving the MLE problem with the regularization term. Statistical guarantees of the MLE objective and correlation kernel are also studied.

**Limitations And Societal Impact:**

- In line 192, it seems that m <= |C| + n since samples in \mathcal{I} can contain points in C_l.
- In Theorem 6, what is the \kappa?


**Main Review:**

This work tries to bridge the gap between continuous DPPs and discrete DPPs and provides some rigorous results. However, some derivation is unnatural and various analysis methods are already known. Moreover, as DPPs are used in many ML applications and mostly they are given by discrete settings, one should justify importance of continuous DPPs rather than discrete DPPs.

The equation (6) comes from penalizing the difference of normalization term as shown in Theorem 1. However, this only considers the error of normalization term and one needs to consider the first term (i.e., numerator in probability of DPP) in equation (5). Does the Theorem 1 similarly validate without identity operator?

In Theorem 6, the authors provide a statistical guarantee of correlation kernel. Is it possible to provide similar result in terms of the likelihood kernel in Algorithm 1?

Overall, it needs to be improved the justification of the contributions of this paper. Also, it would be better to provide more concrete results (e.g., error estimation) rather than visualization.

=========================================================================

I have read authors response and other reviews and noticed there was a misunderstanding regarding continuous DPPs. Authors clarified this in their response. I update my score.

**Time Spent Reviewing:**

5

---

> ### Author Response · Authors · 2021-08-10
> **Answer to Reviewer dbvc**
>
> Thank you for your review.
> > However, some derivation is unnatural and various analysis methods are already known.
>
> We respectfully disagree with the qualification of "unnatural" and the fact that our work has little contribution over known analysis methods. While the individual techniques that we use, like sampling approximations and the representer theorem of [Marteau-Ferey et al., 2020], are indeed borrowed from a rich literature on both randomized numerical algebra and RKHSs, chaining them together and applying them to nonparametric DPP inference is novel, to our knowledge and as noted by the two other reviewers. Besides, we had to develop new variants of the aforementioned techniques, which could be interesting per se, like the sampling approximation to Fredholm determinants of Theorem 1.
> > The equation (6) comes from penalizing the difference of normalization term as shown in Theorem 1. However, this only considers the error of normalization term and one needs to consider the first term (i.e., numerator in probability of DPP) in equation (5).
>
> There may be a misunderstading here. In Theorem 1, we describe the error (w.h.p.) incurred if the second term in (5) is discretized. The first term in (5) involves the determinant of a finite matrix, and therefore does not yield any discretization error.
> > Does the Theorem 1 similarly validate without identity operator?
>
> Please correct us if we misread the question, but we understand that the reviewer may be unfamiliar with Fredholm determinants. In short, extending the notion of determinant from matrices to operators is not straightfoward. The so-called Fredholm determinant $\det(\mathbb{I} + B)$ of a trace-class endomorphism $B$ of a Hilbert space can be defined in a variety of ways naturally extending the usual determinant, and has been studied in detail in operator theory. But it is not clear how to properly define a similar determinant without the identity term. As a short and computationally-minded introduction to these problems, we refer to [Bornemann, On the numerical evaluation of Fredholm determinants, SIAM Mathematics of Computation, 2020].
> > In Theorem 6, the authors provide a statistical guarantee of correlation kernel. Is it possible to provide similar result in terms of the likelihood kernel in Algorithm 1?
>
> Good question. The finite-rank operator $V\mathbf{B}_\star V^*$ in Corollary 5 is our approximation to the likelihood kernel. The corollary guarantees that its log-likelihood is close to that of the actual MLE, which justifies our whole pipeline. Proving also that the two optimizers are close, i.e. that our approximate likelihood kernel is close to the MLE in some appropriate norm, is a natural (but technical) next step, which we have left to future work.
> > In line 192, it seems that $m \leq |\mathcal{C}| + n$ since samples in $\mathcal{I}$ can contain points in $\mathcal{C}_l$.
>
> Indeed, but since $\mathcal{I}$ is sampled from the uniform measure on the domain, the probability that $m = |\mathcal{C}| + n$ is one. We will clarify this.
> >  In Theorem 6, what is the $\kappa$?
>
> The constant $\kappa^2$ is defined L62 as a uniform bound on the diagonal of the RKHS kernel. We will refer to the definition in the statement of Theorem 6.

---

> > ### Comment · Reviewer_dbvc · 2021-08-17
> > **Follow-up Comment**
> >
> > I would like to thank the authors for the detailed answers to my questions. There was a misunderstanding regarding the continuous DPP and the Fredholm determinants. The reviewers comments clarified these concepts as well as other questions.
> >
> > Based on their response and comments from other reviewers, I agree that this work answers new results of nonparametric estimation of continuous DPPs which is not so studied in prior and can be also good hints for learning discrete DPPs. Besides, the result itself is technically sound and fairly well written. Following these reasons, I raise the score to 7.

---

### Decision · Program_Chairs · 2021-09-27

**Decision:**

Accept (Poster)

**Comment:**

This paper studies nonparametric maximum likelihood estimation for continuous DPPs. In contrast to discrete DPPs, for continuous ones there are many open questions and new challenges. The main results are based on recent representer theorems for continuous DPPs as nonnegative functions in an RKHS. The paper uses this machinery to develop a new fixed point algorithm which is guaranteed to increase the MLE objective after each step. Furthermore they give strong statistical guarantees – bounds on the error of their estimator in the appropriate metric, depending on the sample complexity. All the reviewers agreed that the results were interesting and original, and that the paper was well-written and accessible.